



# Statistical impact of wind-speed ramp events on turbines, via observations and coupled fluid-dynamic and aeroelastic simulations

Mark Kelly[1], Søren Juhl Andersen[2], Ásta Hannesdóttir[1]

[1]Department of Wind Energy, Danish Technical University, Risø Lab/Campus, Roskilde 4000, Denmark
[2]Department of Wind Energy, Danish Technical University, Lyngby 2800, Denmark

*Correspondence to*: Mark Kelly (MKEL@dtu.dk)

**Abstract.** Via 11 years of high-frequency measurements, we calculated the probability space of expected offshore wind speed ramps, recasting it compactly in terms of relevant load-driving quantities for horizontal-axis wind turbines. A statistical ensemble of events in reduced ramp-parameter space (ramp acceleration, mean speed after ramp, upper-level shear) was

created, to capture the variability of ramp parameters and also allow connection of such to ramp-driven loads. Constrained Mann-model (CMM) turbulence simulations coupled to an aero-elastic model were made for each ensemble member, for a single turbine. Ramp acceleration was found to dominate the maxima of thrust-associated loads, with a ramp-induced increase of 45–50% for blade-root flap-wise bending moment and tower base fore-aft moment, plus ~3% per 0.1 m s$^{-2}$ of bulk ramp acceleration magnitude.

The ensemble of ramp events from the CMM was also embedded in large-eddy simulation (LES) of a wind farm consisting of rows of nine turbines. The LES uses actuator-line modelling for the turbines and is coupled to the aero-elastic model. The LES results indicate that the ramps, and the mean acceleration associated with them, tend to persist through farm. Depending on the ramp acceleration, ramps crossing rated speed lead to maximum loads, which are nearly constant for the third row and further downwind. Where rated power is not achieved, the loads primarily depend on wind speed; as mean winds weaken

within the farm, ramps can again have $U<V_{rated}$. This leads to higher loads than pre-ramp conditions, with the distance where loads begin to increase depending on inflow $U_{max}$ relative to $V_{rated}$. For the ramps considered here, the effect of turbulence on loads is found to be small relative to ramp amplitude that causes $V_{rated}$ to be exceeded, but for ramps with $U_{after}<V_{rated}$, the combination of ramp and turbulence can cause load maxima. The same sensitivity of loads to acceleration is found in both the the CMM-aeroelastic simulations and the coupled LES.

## 1 Introduction

The passage of ramp-like events, whereby wind speed increases significantly over a span of seconds or minutes, can significantly affect the performance of megawatt-scale wind turbines, in terms of loads as well as power production. These events are often associated with the passage of cold fronts (e.g. Musilek & Li, 2011), but are also caused by a number of other mechanisms, depending on the surroundings (Gallego-Castillo *et al.,* 2015; Hannesdóttir & Kelly, 2019). Wind ramps can



persist through entire wind farms (as we also show below)—more so than turbulent fluctuations, which are limited in scale
and become affected by the turbines themselves (Andersen et al., 2017b).

The basic ramp quantities directly associated with wind-speed ramp events are the rise time ($\Delta t$) and ramp amplitude ($\Delta U$).
In addition to these, a number of observable atmospheric flow quantities can affect turbine loads and performance during the
passage of such events; the ramp acceleration $\Delta U/\Delta t$ can affect the loads and production more significantly than $\Delta U$ or $\Delta t$
alone (e.g. Hannesdóttir, 2018; Hannesdóttir et al., 2021).  The primary observable non-ramp quantity, which is expected to
affect a windfarm's response to ramps, is the above-rotor shear connected with the capping inversion of the atmospheric
boundary layer (Abkar & Porté-Agel, 2013; Kelly et al., 2019a); it is an indicator of how much momentum can mix downward
into the farm, and presumably affect the ramp decay (e.g. Porté-Agel et al., 2020).  Although the standard deviation of
streamwise velocity fluctuations ($\sigma_u$) and turbulence length scale ($L_t$) generally affect turbine loads and performance, for
ramp-like events Hannesdóttir et al. (2019) found them to be secondary compared to ramp-associated quantities.  Thus we are
considering the effects on turbine loads and power for a given rated speed $V_{\text{rated}}$, in terms of the variable space consisting of
$\Delta U, \Delta t, (dU/dz)_{\text{top}}, U_{\text{before}}, \sigma_u$, and $L_t$.  More specifically, we aim to connect the variability in turbine loads to the long-
term statistics of ramps; i.e., to find the statistical effect of wind ramps on wind farms.


There are a number of limitations within commonly used models and observations, which motivate the methods we will use
in this study. Within weather-forecasting models, the inability of planetary-boundary layer schemes to represent various
physical processes giving rise to ramps, have limited the ability of the former to predict ramp-like events (Jahn et al., 2017).
Regarding observations, DeMarco & Basu (2018) looked at statistics primarily based on 10-minute averages, with limited
analysis based on 1-minute averages from a site in the lee of steep mountains; others have also considered 10-minute mean
statistics, but these do not reliably capture the accelerations (or variability) inherent in ramps—nor permit systematic
connection of ramp characteristics with turbine loads, due to the shorter time scales involved (e.g. Alcayaga, 2017; Dimitrov
& Natarajan, 2017).  Thus we examine observational data with sampling rates high enough to adequately characterize ramp-
like events (1 Hz), and employ models which can also resolve velocity fluctuations to such a fine timescale as well as resolving
velocity fields with a resolution significantly smaller than turbine blades.

We first report on the probabilistic characterization of wind speed ramps, and then describe two associated model chains
developed and employed to simulate the propagation of ramps through a windfarm; this is followed by analysis of the
modelling results for relevant turbine loads.  The probabilistic characterization involves reduction of atmospheric quantities to
a more compact and universal space, as well as creation of a statistically representative ensemble of events which can be
simulated in the two model chains. The two model chains are both driven by Mann-model turbulence (Mann, 1994, 1998),
with the synthesized turbulence constrained (Dimitrov & Natarajan, 2017) to include wind speed ramps (Hannesdóttir et
al., 2019). The simpler model 'chain' also has the aero-elastic model Flex5 for a single turbine; the other embeds constrained



ramp simulations within large-eddy simulation (LES) of a wind farm with actuator line modelling (Sørensen & Shen, 2002)
coupled to Flex5.

## 2 Statistical characterization for model-chain simulations of wind-speed ramps

In order to obtain statistics describing offshore wind speed ramps, we analysed the longest timeseries of high-frequency wind data available at common turbine hub heights: 11 years of 1 Hz wind velocity data from the Høvsøre turbine test center on the
western coast of Denmark (Peña *et al.*, 2016). Using the streamwise velocity at100 m height, for the dominant winds crossing the coastline, we are able to effectively obtain offshore statistics at and above this height from Høvsøre (see also e.g. Berg *et al.*, 2015). To detect ramp events, 10-minute records with the largest variances relative to turbulence strength are selected; the top 0.1% of values of $\sigma_u / (\sigma_{u,hpf} + 1\,\text{m/s})$ are found, where $\sigma_{u,hpf}$ is the turbulent (high-pass filtered) part of $\sigma_u$—as in Hannesdóttir & Kelly (2019)[1].   Keeping events where the wind speed is increasing, we identify 216 wind ramp events.
Distributions of $\Delta U$ and $\Delta t$, which are obtained by the ramp-detection method for each ramp, are shown in Figure 1.

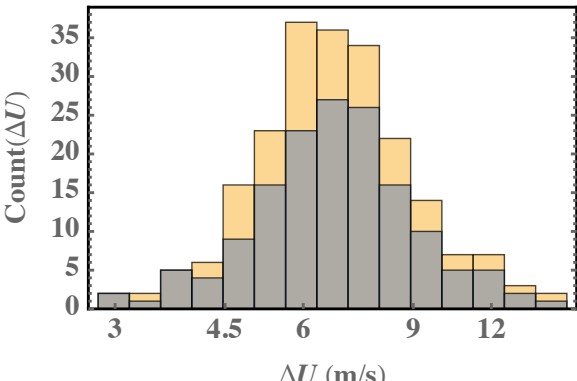
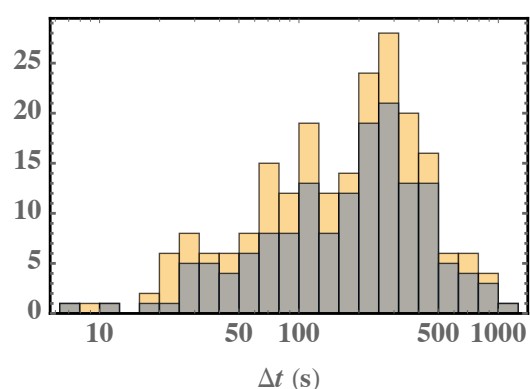

**Figure 1: Distributions of ramp amplitude and rise-time. Yellow: all events; blue/dark: excluding events which exceed cut-out or start above rated speed.**


One consideration we add is whether the ramps begin below rated wind speed and subsequently cross over it—as well as whether the wind speed exceeds the turbine cut-out speed ($V_{cut}$) for a given ramp.  Our aeroelastic modelling employs a NEG/Micon (Vestas) NM80 turbine (Aagard Madsen *et al.*, 2010; Galinos & Larsen, 2015), with upscaled rated power of 2.75MW at $V_{rated} = 14$ m/s and cutout speed $V_{cut} = 25$ m/s (Andersen *et al.*, 2017a)[2].  The dominant ramp effects on power

---

[1] This method finds the strongest ramps relative to turbulence, while also rejecting cases with weak turbulence where $\sigma_{u,hpf}$ is appreciably smaller than 1 m/s. Further explanation is included in the next section, and more details can be found in Hannesdóttir & Kelly (2019).

[2] The choice of turbine was due to the supporting project applying the Rødsand II wind farm (e.g. Nygaard & Hansen, 2016).



and loads occur during normal operation (Hannesdóttir *et al.*, 2019), and ramps have relatively smaller effect on operation at speeds above $V_{rated}$ because the blades have already pitched; thus we exclude ramp cases where the starting speed is above $V_{rated}$ or where the ramp-end speed is above $V_{cut}$. The effect of this can be seen in Figure 1; while the frequency of occurrence is slightly reduced, the shapes of both $P(\Delta U)$ and $P(\Delta t)$ are essentially unaffected. We further note that both distributions (particularly the rise time) are better represented in log-space, with the distribution of ramp amplitude appearing to be log-

normal.

Similar to the ramp magnitude and duration, the distribution of acceleration is also seen more conveniently in logarithmic space; this is shown in Figure 2. Unlike $P(\Delta U)$ and $P(\Delta t)$, the shape of the ramp-acceleration distribution $P(\Delta U/\Delta t)$ is affected by rejection of events which exceed cut-out turbine speed and start above rated speed. From Figure 2 one can see the

smallest accelerations being filtered out; those below 0.01 m/s² are roughly halved. These do not affect the turbine, since turbulent accelerations override the ramp for such small $\Delta U/\Delta t$; furthermore, the high-acceleration tail remains essentially unchanged by the filtering, as seen in the figure.

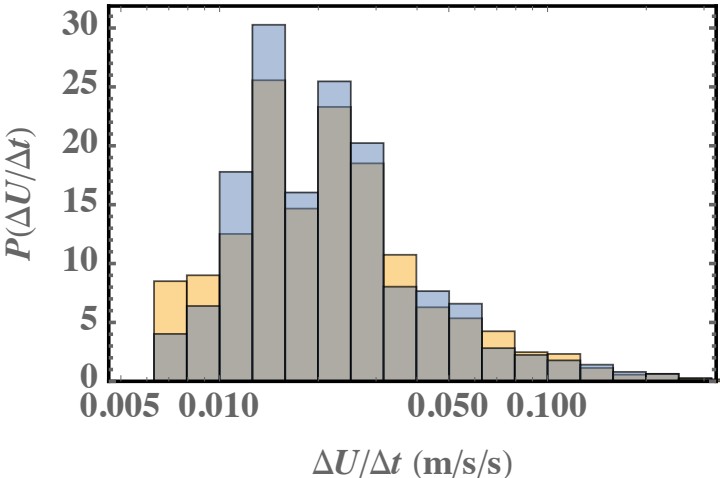

**Figure 2: Probability density function of (mean) ramp accelerations, with logarithmic axis for $\Delta U/\Delta t$. Yellow: all events; blue/dark: excluding events which exceed cut-out or start above rated speed (as in Figure 1).**

The upper-rotor shear is also calculated, using the anemometer at 160 m along with that at 100 m. Although some ramps can be tilted in the streamwise direction (i.e., closer to the ground the ramp arrives later) and possess a transient shear associated

with such tilt, this has a relatively small effect on the loads considered (Hannesdóttir *et al.*, 2017)[3], and is beyond the scope of

---

[3] The transient shear was shown in Hannesdóttir *et al.* (2017) to induce tower-top yaw (and possibly tilt) moments stronger than those induced by the design-load case prescription for transient shear (DLC1.5) in the IEC 61400-1 standard, but these magnitudes do not exceed



the current study. The shear ($dU/dz$) tends to be different before and after ramps, especially because most of these events are related to the passage of fronts. As mentioned above, the shear above the rotor is the most readily measured 'external' factor that can be used to diagnose downward entrainment of momentum and turbulent mixing into the farm. It is connected with atmospheric stability (Kelly *et al.*, 2014), and particularly influenced by the capping temperature inversion and thus the depth

of the atmospheric boundary layer (Kelly *et al.*, 2019a). Distributions of the free-stream shear across the rotor (calculated as $U_{160m} - U_{60m}$) and the upper/above-rotor shear ($U_{160m} - U_{100m}$), both before and after ramp passage, are shown in Figure 3. We examine $dU/dz$ instead of the shear *exponent* ($\alpha = d \ln U / d \ln z$) because $dU/dz$ is directly involved in the momentum (entrainment) flux.[4] Figure 3 also shows that the shear before the ramps is on average slightly larger than after the passage of a ramp event, and the upper-rotor $dU/dz$ is smaller than the full-rotor shear. We note the shear before and after ramp events

is unrelated, with an increase or decrease possible; the distribution of the difference between shear before and after (not shown) is centered around 0 but has a width comparable to the shear distributions themselves.

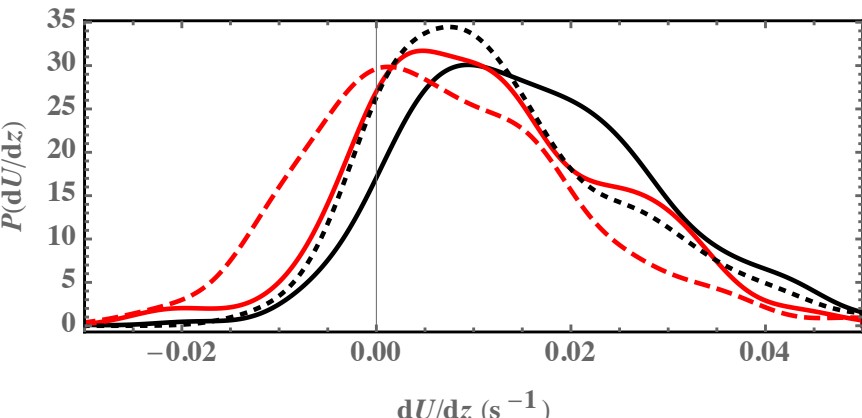

**Figure 3: Probability distribution of shear in the periods before (black) and after (red) the detected ramp events. Solid denotes**
**whole-rotor shear (as $U_{160m}$–$U_{60m}$); dotted/dashed denotes upper-rotor shear ($U_{160m}$–$U_{100m}$).**

We do not directly address the strength of turbulent fluctuations ($\sigma_u$), i.e. the turbine response statistics related to variability in $\sigma_u$, because the ramp amplitudes and associated accelerations are generally large enough to be more significant than such background turbulence. This is supported by comparison of Figure 1 with distributions of $\sigma_u$ shown in Figure 4, where each

---

the IEC extreme turbulence prescription (DLC1.3). Here we consider primarily the tower base fore-aft moment and blade root flap-wise bending moments, which for the observed ramps exceed DLC 1.3 in the 61400-1 (see also Hannesdóttir *et al.*, 2019).

[4] We remind that $dU/dz$ tends to be well-correlated with (at least monotonic in) the momentum flux $\langle uw \rangle$; the latter is typically parameterized as proportional to the former via first-order closure and mixing-length turbulence models (e.g. Panofsky & Dutton, 1984). Production of turbulent kinetic energy (TKE) is also proportional to $dU/dz$, The vertical (entrainment) flux of *mean* kinetic energy ($\sim \rho U^2/2$) can be nonlinear in the shear, but is yet more sensitive to $dU/dz$ (see e.g. Ch. 5 of Wyngaard, 2010). Comparing with Kelly *et al.* (2014), and noting the logarithmic character of $\alpha$, one can see these fluxes are not as directly related and are less sensitive to $\alpha$. However, use of $\alpha$ for flux parameterizations can certainly be explored further.





$\sigma_u$ 'sample' is calculated over the full (10-minute) period corresponding to a given ramp, and the probability density function (PDF) is the collection of all ramp-period samples. The portion of wind variation due to turbulence ($\sigma_{u,\text{hpf}}$), apart from the ramp, is also shown; this is calculated using a second-order high-pass Butterworth filter and filter frequency of $f_c = U/L_c$, where $U$ is the 10-minute mean wind speed and $L_c = 2$ km as in Hannesdóttir and Kelly (2019). In Figure 4 one sees that the turbulence variation $\sigma_{u,\text{hpf}}$ is small compared to the ramp amplitudes $\Delta U$ and the $\sigma_u$ associated with the ramps: the peak of

$P(\sigma_{u,\text{hpf}})$ is at less than 1 m/s, whereas the peak of $P(\sigma_u)$ is ~2.5 m/s, and the peak of $P(\Delta U)$ is about 7 m/s with $\Delta U$ ranging from 3–15 m/s for the events considered. One also sees that rejection of the cases exceeding cut-out does not affect $\sigma_u$, though it is tied to a slight reduction in the turbulence strength $\sigma_{u,\text{hpf}}$. However, this is not significant, given that the ramps dominate the inflow to the turbines.

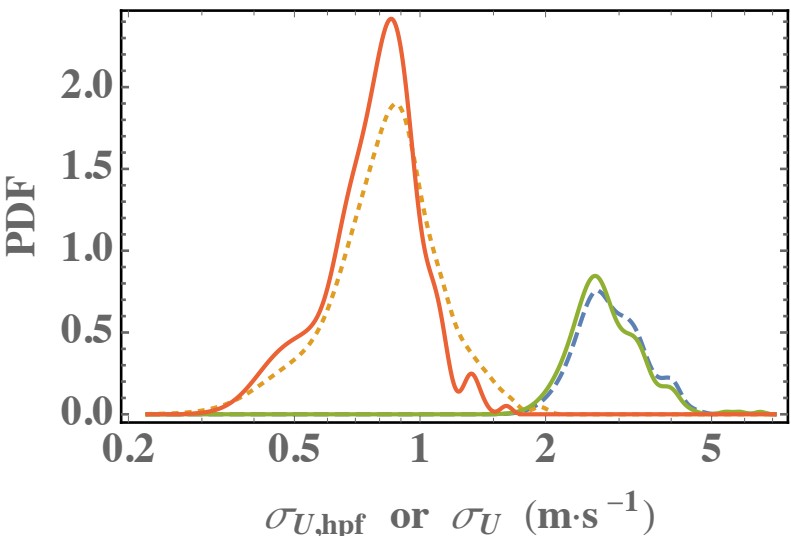


**Figure 4: Probability density functions of standard deviation of wind speed over all ramp periods. Dashed-blue is unfiltered $\sigma_u$ over all ramps; dashed-orange is high-pass filtered turbulence ($\sigma_{u,\text{hpf}}$); solid green is $\sigma_u$ excluding events exceeding $V_{\text{cut}}$ or starting above $V_{\text{rated}}$; solid red is $\sigma_{u,\text{hpf}}$ also excluding such events.**

**2.1 Joint statistics: practical and systematic event characterization**

In order to investigate the relevant statistical space describing the inflow encountered by turbines during ramp events, we look deeper than the *marginal* distributions shown above. An initial picture of the ramp event probability space is given by Figure 5, which displays each ramp as a point $\{\Delta t, \Delta U\}$. Due to the consideration of $V_{\text{rated}}$ and $V_{\text{cut}}$ it is useful to include the wind speed; the figure also displays $U_{\text{before}}$ for each ramp-like event found. Figure 5 indicates a concentration of most likely rise

times and amplitudes around $\{\Delta t, \Delta U\} \approx .\{7–8$ m/s, $300–400$ s$\}$, which can also be seen in the joint distribution of $\Delta t$ and $\Delta U$





(not shown) and consistent with the distributions $P(\Delta t)$ and $P(\Delta U)$ shown earlier in Figure 1. Further, one can see that ramps are more often preceded by relatively strong winds; this is consistent with passage of cold fronts in mid-latitude areas.

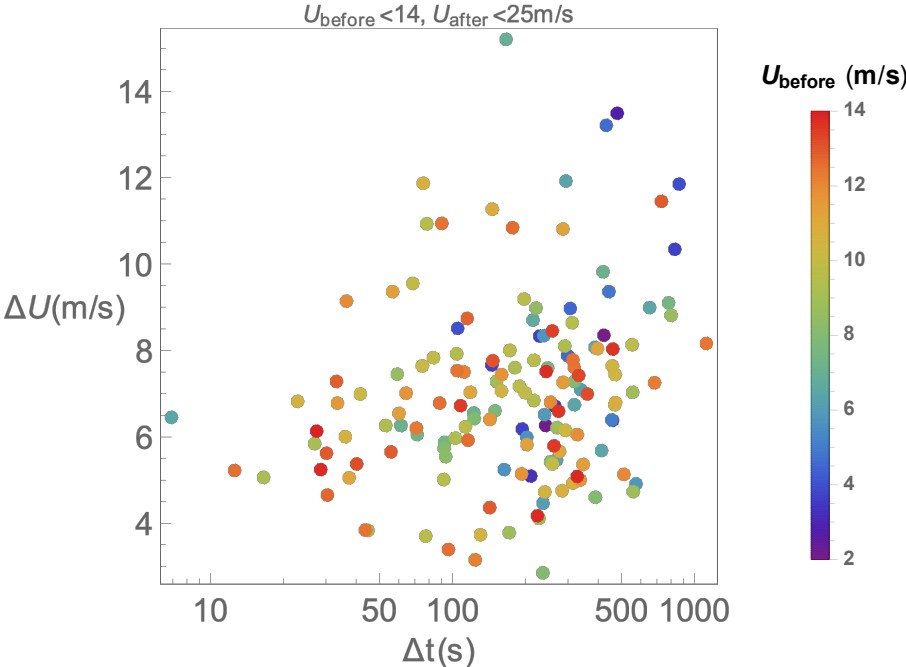

**Figure 5: Ramp event occurrence, in terms of $\{\Delta t, \Delta U, U_{\text{before}}\}$.**

For a given turbine, the rated speed is expected to have an impact on its response to ramps. So in Figure 6a we show the joint distribution of ramp amplitudes and pre-ramp wind speeds, $P(\Delta U, U_{\text{before}})$, as well as indicating which events cross $V_{\text{rated}}$. Figure 6b shows the distributions of mean wind-speeds before and after ramp events. From this $P(\Delta U, U_{\text{before}})$ and Figure 1

we note that the most likely ramp events with amplitudes below the peak of $P(\Delta U)$ tend to occur with initial speeds larger than what the peak of the simple marginal distribution $P(U_{\text{before}})$ would seem to imply; i.e. there is a joint trend where smaller ramp amplitudes tend to occur with larger pre-ramp wind speeds. One further sees in Figure 6a the number and distribution of events which involve wind speed crossing $V_{\text{rated}}$, indicated by the yellow line; a fraction of ramps (~1/6) have $U_{\text{after}} < V_{\text{rated}}$. Figure 6a also indicates the small number of rejected events (~5%) exceeding turbine cut-out speed, which

lay above the blue line.



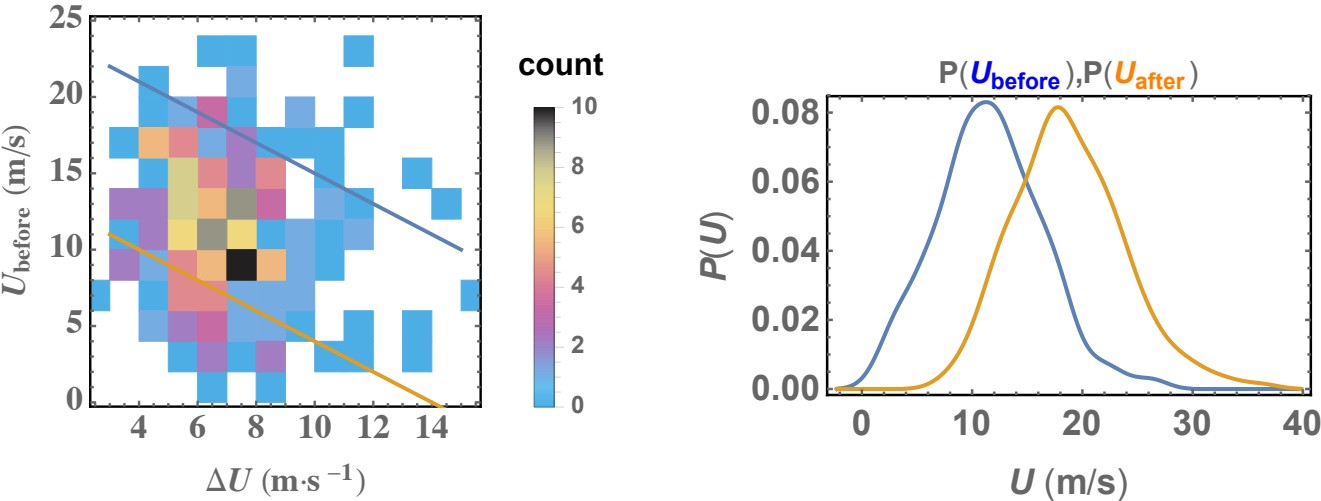

**Figure 6.** Left: Joint distribution of ramp amplitude and speed before ramp; points above blue line are above cutout (rejected), and points below yellow line do not cross from below to above $V_{\text{rated}}$. Right: distribution of speed before and after ramp events.


As mentioned above, in previous related work with aeroelastic simulations, wind turbines have shown more sensitivity to ramp acceleration ($\Delta U/\Delta t$) than to ramp amplitude ($\Delta U$). But aside from its contribution to $\Delta U/\Delta t$, the ramp amplitude can have primary significance for events which do not exceed $V_{\text{rated}}$ (points falling below the yellow line in Figure 6), as we will see in the next section. The strongest ramp accelerations appear to be correlated with the wind speed before ramp passage; this is

demonstrated by Figure 7, which displays $P(\Delta U/\Delta t, U_{\text{before}})$. For the largest bulk accelerations, one sees that $\ln(\Delta U/\Delta t)$ roughly follows $U_{\text{before}}$.

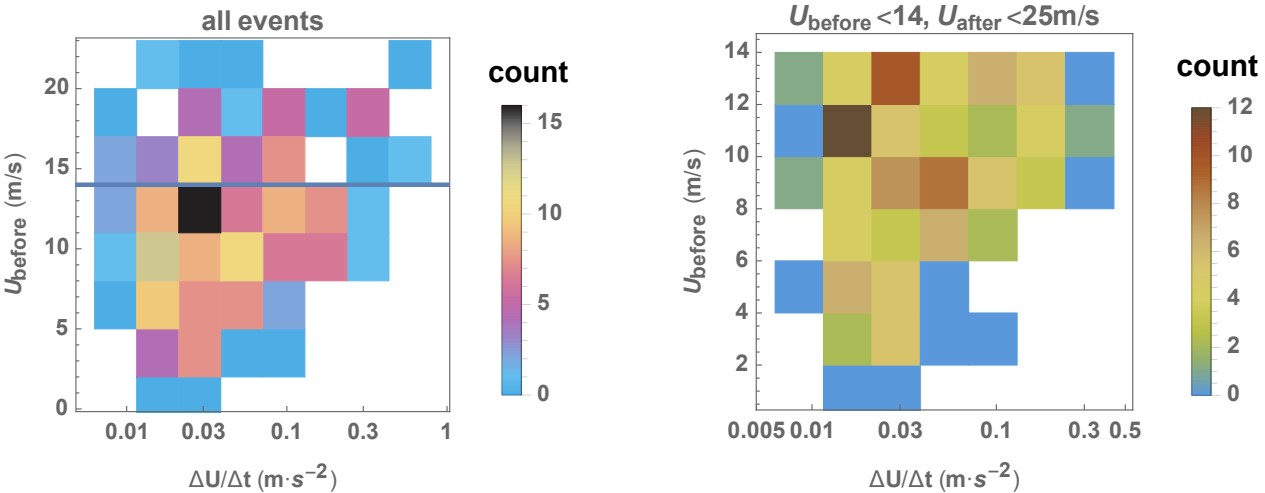

**Figure 7:** Distribution of accelerations per wind speed before ramp. Left: all events, where horizontal blue line shows lower limit of
rejected events; right: events considered here.



Given the primary impact of ramp acceleration ($\Delta U / \Delta t$) on lone turbines and the influence of above-rotor shear on mixing within wind farms, we examine their joint behavior. Figure 8 shows each event's acceleration and the upper-rotor shear after passage, as well as the pre-ramp speed. Recalling the essentially offshore conditions (again the ramps are from the west and

the speed at 100 m height is unaffected by the thin strip of land between mast and coastline), the relatively small shear matches previous observations, with a moderately skewed distribution for $\alpha$ (Kelly *et al.*, 2014) and associated $P(\Delta U / \Delta z)$. The range of shear appears wider for the most commonly occurring accelerations ($\Delta U / \Delta t \sim 0.02$–$0.04$ m/s$^2$)—particularly for events with lower speeds—though there is little evidence of shear correlating with the bulk ramp acceleration. For 'stronger' ramps, i.e. with the highest accelerations, the shear is weaker, which in part justifies use of neutral conditions in the simulations;

e.g. for $\Delta U / \Delta t > 0.2$ m/s$^2$, basically $|\Delta U / \Delta z|_{\mathrm{upper}} < 0.01$ s$^{-1}$. The character of $P(\Delta U / \Delta t, \Delta U / \Delta z, U_{\mathrm{before}})$, evinced by Figure 8, motivates our choice of event ensemble for simulations shown in the next section.

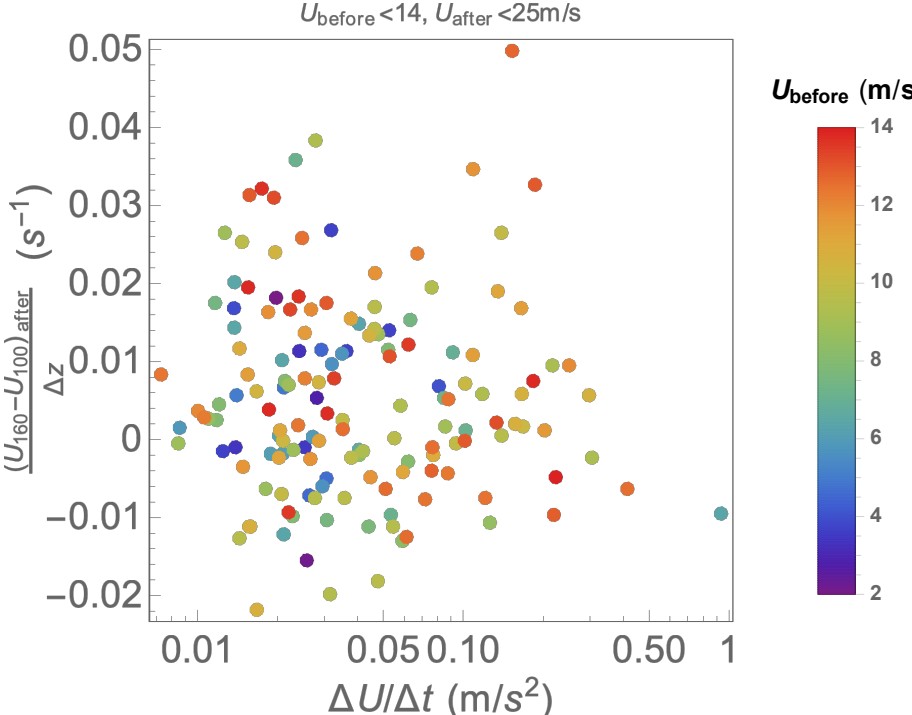

**Figure 8: Detected ramp events: upper-level shear after event, ramp acceleration, and wind speed before ramp.**






## 2.2 Ensemble of ramp events for coupled simulations

Due to the computational demands of the model chain used, an ensemble of 8 members was created based on the joint statistics presented in the previous section. Each ensemble member corresponds to one model-chain simulation. The model-chain starts with a constrained turbulence simulation, which employs the Mann-model (Mann, 1994, 1998, 2001); output from the

constrained Mann-model (CMM) is used to drive the coupled LES and aeroelastic models, as detailed in the next section.

The ensemble members are chosen to cover the relevant load-driving parameters: the ramp acceleration, with the latter dictated by the ramp duration and amplitude; the pre-ramp hub-height wind speed (shown previously in Figure 8); and the above-rotor shear. Based on the marginal PDF of background turbulence strength $\sigma_{u,\mathrm{hpf}}$ (Figure 4) and joint distributions of it with the

other parameters (not shown), a single representative value for $\sigma_{u,\mathrm{hpf}}$ was chosen. Since we are not investigating sensitivity to $\sigma_{u,\mathrm{hpf}}$, and because it (and its effect) is small compared to $\Delta U$, we choose a value equal to the observed mean, which is approximately equal to the mean within the $\{\Delta U/\Delta t, \Delta U_{\mathrm{upper}}/\Delta z\}$ space considered. Two values of $\Delta U$ and several values of $\Delta t$, corresponding to three significant accelerations $\Delta U/\Delta t \geq 0.05$ m/s$^2$, were chosen; this was done in such a way as to [1] cover the most populated part of the statistical space and [2] facilitate estimation of the sensitivity of turbine response to

wind speed ramps, particularly via $\Delta U/\Delta t$.

Similarly, three values of pre-ramp shear were chosen, to investigate dependence of ramp propagation through the farm, and sensitivity. A representative value of the turbulence length scale $L_{\mathrm{MM}}$, is also needed for constrained turbulence simulations. Since $L_{\mathrm{MM}}$ is not expected to be a significant driver of loads due to the dominance of the ramps (it has less influence than

$\sigma_{u,\mathrm{hpf}}$, as shown in e.g. Dimitrov *et al.*, 2018), it was calculated as $L_{MM} = \sigma_{u,\mathrm{hpf}}/(dU_{\mathrm{before}}/dz)_{\mathrm{upper}}$ following Kelly (2019)[5]. Table 1 shows the ensemble-members and chosen characteristics.

---

[5]In the zero-shear cases, $L_{\mathrm{MM}}$=200 m was assigned based on the spectral length scales diagnosed in Kelly (2019) from these same data, under the condition d$U$/d$z \simeq 0$



**Table 1. Ensemble of ramp events: parameters chosen.**

| Case | $(dU/dz)_{\text{before}}$ | $\Delta U_{\text{ramp}}$ | $\Delta t$ | $\Delta U_{\text{ramp}}/\Delta t$ | $U_{\text{before}}$ | $\sigma_{u,\text{hpf}}$ | $L_{MM}$ |
|------|---------------------------|--------------------------|------------|-----------------------------------|---------------------|-------------------------|----------|
| 1 | 0.02 s$^{-1}$ | 9 m/s | 90 s | 0.1 m/s$^2$ | 10 m/s | 0.9 m/s | 45 m |
| 2 | 0.02 s$^{-1}$ | 6 m/s | 60 s | 0.1 m/s$^2$ | 10 m/s | " | 45 m |
| 3 | 0 s$^{-1}$ | 9 m/s | 90 s | 0.1 m/s$^2$ | 10 m/s | " | 200 m |
| 4 | 0.02 s$^{-1}$ | 9 m/s | 180 s | 0.05 m/s$^2$ | 10 m/s | " | 45 m |
| 5 | 0 s$^{-1}$ | 9 m/s | 180 s | 0.05 m/s$^2$ | 10 m/s | " | 200 m |
| 6 | 0.01 s$^{-1}$ | 6 m/s | 30 s | 0.2 m/s$^2$ | 10 m/s | " | 90 m |
| 7 | 0 s$^{-1}$ | 6 m/s | 240 s | 0.025 m/s$^2$ | 6 m/s | " | 200 m |
| 8 | 0.01 s$^{-1}$ | 6 m/s | 120 s | 0.05 m/s$^2$ | 6 m/s | " | 90 m |


## 3 Constrained turbulence simulation with ramps

The ensemble members defined according to the specified parameters $\{\Delta U, \Delta t, (\Delta U_{\text{before}}/\Delta z)_{\text{upper}}, U_{\text{before}}, \sigma_{u,\text{hpf}}, L_{MM}\}$, shown in Table 1, were each used to generate a constrained turbulence simulation with Mann-model background turbulence (Dimitrov & Natarajan, 2017)—as done by Hannesdóttir *et al.* (2019) for wind ramps. The duration of the

simulations was 1550 s, producing a three-dimensional turbulence "box" of atmospheric turbulence velocity field including the ramp for each member. The ramps begin 800 s after simulation start to ensure fully developed flow through the farm without initial transients, and are of sufficient duration to include both the ramp and at least 500 s of high-wind turbulence after the ramp (as in the observations).

The simulations are stochastic, including turbulence generated by the Mann model for atmospheric turbulence (c.f. Mann, 1994,1998; IEC 61400-1, 2019), so the resulting ramps simulated by LES are not exactly as specified in Table 1. The superposed turbulence with the ramps can cause deviations in wind speed, which may either change the duration or amplitude of an event. This is shown in Table 2, which presents the diagnosed ramp parameters from the ensemble of simulations. For example, the ramps in cases 2 and 4 are prolonged, while case 5 and 7 have shortened ramps compared to

the durations chosen. The resultant accelerations are affected, and diagnosed in two ways: the mean $\langle \partial u/\partial t \rangle$ of the accelerations calculated via first-order finite difference over each ramp duration (with $\partial t$ corresponding to 0.04 s), and the bulk ramp value $\Delta U/\Delta t$ (again where $\Delta U = \langle U_{\text{after}} \rangle - \langle U_{\text{before}} \rangle$ and $\Delta t$ is ramp duration). The table includes each, with



$\langle \partial u/\partial t \rangle$ reflecting the nonlinear stochastic aspect and effect on the simulated ensemble. Case 2 has a reduced acceleration, while cases 5 and 7 have larger accelerations than prescribed. The bulk acceleration $\Delta U/\Delta t$ is closer to prescribed accelerations than the average of 'instantaneous' accelerations $\langle \partial u/\partial t \rangle$, because the latter includes more effects of simulated turbulence. Further, because the latter may also be sensitive to temporal resolution of the data[6], we refer hereafter to the bulk value $\Delta U/\Delta t$ as the diagnosed ramp acceleration; because we have prescribed the before- and after-ramp speeds and simulated well beyond 10 minutes duration, this bulk acceleration is also equivalent to that found via the detection algorithm used to identify the ramp events in the original measured data.

**Table 2. Diagnosed parameters from ensemble of constrained (CMM) simulations; ramp acceleration is shown both as average acceleration over the ramp event ($\langle \partial u/\partial t \rangle$ including turbulence), and based on ratio of $[\langle U_{\text{after}} \rangle - \langle U_{\text{before}} \rangle]$ to ramp duration $\Delta t$.**

| Case | $\Delta U$ | $\Delta t$ | $\langle \partial u/\partial t \rangle$ | $\Delta U/\Delta t$ | $U_{\text{before}}$ | $\sigma_{u,\text{hpf (before)}}$ |
|------|-----------|-----------|------------------|-------------|------------|-----------------|
| 1 | 10.4 m/s | 90 s | 0.12 m/s$^2$ | 0.12 m/s$^2$ | 11.9 m/s | 0.83 m/s |
| 2 | 7.2 m/s | ~120 s | 0.06 m/s$^2$ | 0.06 m/s$^2$ | 12.3 m/s | 0.93 m/s |
| 3 | 9.2 m/s | 90 s | 0.07 m/s$^2$ | 0.10 m/s$^2$ | 10.0 m/s | 1.00 m/s |
| 4 | 10.2 m/s | 240 s | 0.04 m/s$^2$ | 0.04 m/s$^2$ | 11.8 m/s | 0.91 m/s |
| 5 | 9.8 m/s | 120 s | 0.06 m/s$^2$ | 0.08 m/s$^2$ | 9.7 m/s | 0.65 m/s |
| 6 | 6.3 m/s | 30 s | 0.28 m/s$^2$ | 0.21 m/s$^2$ | 11.0 m/s | 0.75 m/s |
| 7 | 6.1 m/s | 60 s | (0.07) m/s$^2$ | 0.10 m/s$^2$ | 5.7 m/s | 0.74 m/s |
| 8 | 6.6 m/s | 120 s | (0.07) m/s$^2$ | 0.05 m/s$^2$ | 6.9 m/s | 0.81 m/s |

Note that although there are minor deviations in several cases from the initial ensemble-member choices, this is permissible, given that the cases with deviations are still representative of the joint space—still falling within the populated regions shown in Figure 8, and allowing estimation of sensitivities as originally planned. The ramps from the CMM simulations are shown in Figure 9, where these timeseries correspond to the speeds at {y,z} of the rotor center.

---

[6] The Mann-model output velocity components and consequent accelerations follow the Kolmogorov spectrum, without the high-frequency noise characteristic of measurements. However, given their $f^{-2/3}$ (and thus $\delta t^{2/3}$) dependence, the acceleration still depends slightly on output time step.



### 3.1 Stand-alone aeroelastic calculations driven by the turbulent ramp simulations

The three-dimensional turbulence timeseries for each case were input into the stand-alone aeroelastic code Flex5 (Øye, 1996).
The turbine model and controller employed in Flex5 correspond to the NM80 (Aagard Madsen *et al.*, 2010), as mentioned
previously; this turbine has a rotor diameter $D = 80$ m and hub height $z_{hub} = 80$ m.  The wind speed at hub height in Flex5 is
255   identical to the input speed, as displayed in Figure 9 for the ramp portion of all cases. The plotted timeseries is smoothed using
a 20 s moving average, to show the wind speed 'experienced' by a single simulated turbine—since it reacts like a low-pass
filter with characteristic timescale of ~20 s or longer (e.g. Frandsen *et al.*, 2008).  As prescribed in Table 1, from Figure 9 (and
Table 2) one can note that cases 7–8 start at a lower wind speed and are not designed to exceed rated wind speed, though case
8 does momentarily exceed $V_{rated}$; however, we note that $\langle U_{after} \rangle$ is 11.8 and 13.5 m/s, respectively, for these two cases.  The
turbine power becomes constant during the ramp in cases 1–6, with maximum loads tending to occur during the ramp (shown
further below).

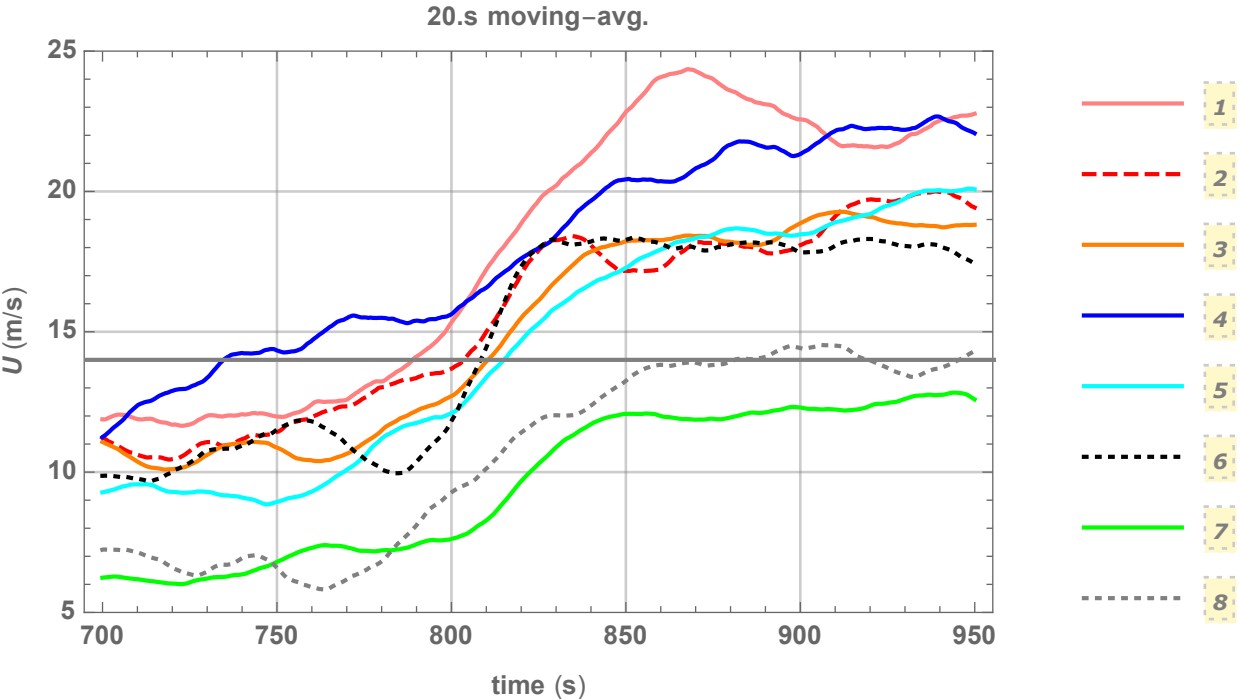

**Figure 9. Wind speed at hub height, for ramps in the stochastic constrained Mann-model simulations (CMM).  Horizontal gray line**
**indicates rated wind speed.  Ensemble member (case) numbers indicated at right.**

From the stand-alone Flex5 simulations, we note the trend of ramp-acceleration dominating the maximum blade root flap-wise
bending moments, as well as the maximum tower-base fore-aft moments (hereafter these two loads are denoted symbolically
by $M_{brfw}$ and $M_{tbfa}$ respectively).  This is shown in Figure 10, which displays max$\{M_{brfw}\}$ and max$\{M_{tbfa}\}$ versus ramp





acceleration for the 7 cases where the speed rises above $V_{rated}$. The acceleration $\Delta U/\Delta t$ is calculated at rotor center (hub height); the maximum from the three blades is used for the blade root-bending moment shown, which was calculated integrating to the first radial computation point in Flex5 (1.24 m from hub). There is some scatter in the results shown due to the spatial variation of turbulence, the shear, and blade positions during the ramp, since $\Delta U/\Delta t$ was calculated at rotor center. However, a trend is evident in the plots, and sensitivity of $M_{brfw}$ and $M_{tbfa}$ to bulk ramp acceleration can be estimated. For the

maximum tower-base fore-aft moment, the sensitivity is roughly 3% of $M_{tbfa}$ per 0.1 m·s$^{-2}$ acceleration due to the ramp, and the sensitivity for maximal flap-wise root bending moment $M_{brfw}$ is approximately the same: 3% per 0.1 m·s$^{-2}$ acceleration. The figure also includes an inset plot where the load was calculated using an averaging time of 4 seconds, which removes scatter and makes the sensitivity yet clearer—with the same slope on this plot as without averaging; it gives the same sensitivity, though the $M_{brfw}$ is simply shifted downward by several percent. In addition to the 3% increase per 0.1 m·s$^{-2}$ of

$\Delta U/\Delta t$, there is an increase of ~45% in $M_{brfw}$ and ~50% in $M_{tbfa}$ for ramp-affected loads, regardless of the ramp amplitude.

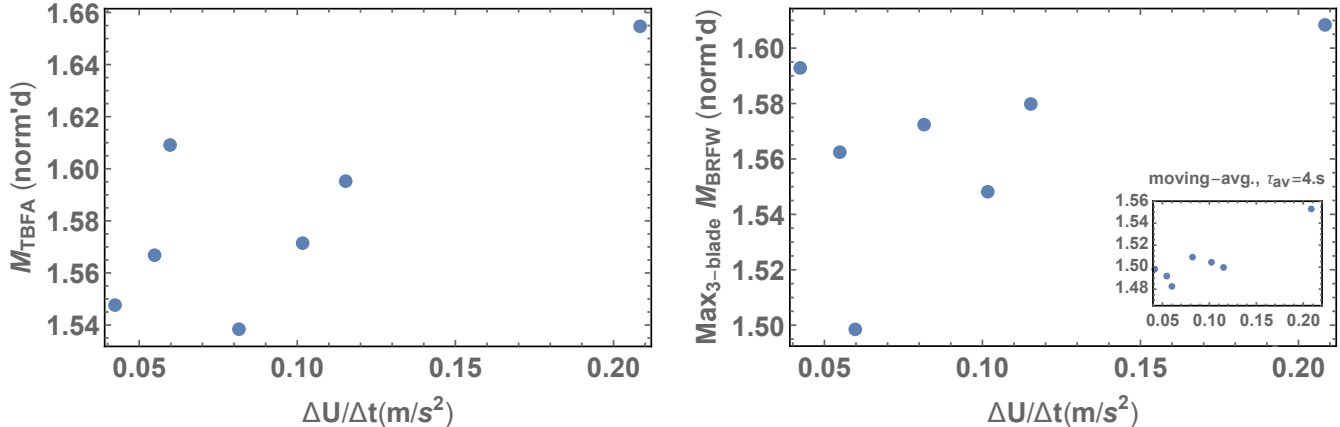

**Figure 10. Tower-base fore-aft moment (left) and blade root flap-wise bending moment (right) versus ramp acceleration, for ramp cases exceeding rated speed, from constrained Mann-turbulence ('stand-alone') simulations. Loads normalized by mean values**
**outside of ramp events, over all cases. Right inset same plot but with averaging time of $\tau_{av} = 100t_s$ =4 s.**

## 4 Coupled large-eddy simulation (LES) of ramps through a simple farm

### 4.1 Large-eddy simulation code

Large-eddy simulations (LES) have been performed using EllipSys3D, which is parallelized Fortran code developed at DTU and the former Risø National Laboratory (Michelsen, 1992; Sørensen, 1995). EllipSys3D solves the discretized incompressible
Navier-Stokes equations in general curvilinear coordinates, using a block-structured finite volume approach. Employing LES implies that resolved (large-scale) motions are solved directly in time and space, while motions and stresses at unresolved (small) turbulent scales are parameterized using a subgrid-scale (SGS) model. The resolution is usually limited by the grid size of the domain, and in the present case is twice the grid spacing, with the 'mixed-scale' SGS parameterization of Ta Phuoc *et*





*al*. (1994) used here. Additionally, body forces are included in the equations and utilized to model the turbines and the turbulent
inflow including ramps, as explained below. See Sørensen *et al*. (2015) for a detailed description of the entire numerical setup.

### 4.1.1 Actuator-line modelling of turbines and aero-elastic coupling

The turbines are modelled using the actuator line method as developed by Sørensen and Shen (2002). The actuator line method
consists of imposing body forces along rotating lines in the computational domain, which corresponds to the aerodynamic
loads on the rotor blades. The body forces are determined through a full coupling to the aero-elastic tool Flex5 (Øye, 1996).
Flex5 computes both the aforementioned forces, as well as blade deflections, based on the instantaneous flow solution along
the rotating lines. Forces and deflections are transferred back to the flow solver. Flex5 also includes a realistic wind turbine
controller, which is particularly important for the current simulations, as the operational regime changes from below rated to
above rated as the ramps propagate through the wind farm.


### 4.1.2 Embedded Body-force implementation to drive the LES

In LES the flow is typically driven via boundary conditions. However, modelling ramps propagating through a domain (which
includes wind turbines) is a particularly challenging task: it violates a fundamental assumption, conservation of mass. The
sudden increase in momentum—particularly the large velocity gradient defining the ramp—can result in an unphysical
acceleration through the simulation domain, due to enforcement of the continuity equation by the pressure solver. Therefore
such persistent transient features *cannot be applied simply on the inlet boundary*. However, use of body forces internally within
the numerical domain facilitates simulation of such flows. The turbulent inflow including the ramp is introduced via body
forces (Gilling *et al.*, 2009), imposed in a plane upstream of the turbine(s) as in Troldborg (2009). If modelling a single turbine,
the plane is often limited in extent. However, as shown in Andersen (2014), limiting the spatial extent of the turbulence plane
can affect the overall mixing in large wind farms.

Here we apply the body-force method with several adaptations, to mitigate the flow degradation within the wind farm and
avoid numerical blockage, which could otherwise occur as an imposed ramp propagates through the simulated domain. Flow
degradation is prevented by ensuring that the imposed 'inflow' covers a large enough area to maintain the ramp throughout
the wind farm; i.e., large-scale compensation for the ramp-induced divergence (via the pressure solver) is kept outside the
central area of interest, such that it has negligible effect within the simulated wind farm. Additionally, the forcing is spatially
tapered with a half-gaussian profile (with scale $1.75D$ in the lateral and $1D$ in the vertical) to reduce anomalous shear, which
would otherwise introduce unintended mixing. Artificial numerical blockage of the imposed ramp inflow is reduced by
utilizing a domain extent of $L_y = 7D$, $L_z = 50D$ that is much larger than the size of the body-forcing area, which has a core area
of $4D \times 2.625D$; this has the additional advantage of distributing any divergence-related compensation over a large volume



such that the induced velocities are small compared to the background speed. The full details of the methodology is presented in Andersen *et al*. (2021).

### 4.1.3 Numerical setup

The numerical domain used for the simulations is 5440m x 560m x 4000m in the streamwise, lateral and vertical directions, respectively. The domain is uniform in the center (where the turbines are placed) and stretched towards the boundaries. The resolution in the center region is 4 m in each direction. There are 1280 x 80 x 64 points in each direction, corresponding to a total of about 6 million cells. This would appear to be relatively coarse for actuator line simulations, but the wake is predominantly governed by $C_T$ (van der Laan *et al*., 2020) and the current resolution gives rise to a difference in $C_T$ of only
~1% (Hodgson et al., 2021).

The flow has been initialized with a power-law wind profile (constant shear exponent), which at hub height corresponds to the vertical velocity gradient given in Table 1. This is also used as the inflow boundary condition. The simulated wind farm contains nine turbines separated in the streamwise direction, by a spacing of seven rotor-diameters (560 m). There are periodic
boundary conditions, so in effect the farm can be considered infinite in the crosswind direction, with a lateral spacing of roughly nine rotor diameters (~700 m) due to the lateral size of the domain. The layout of the simulated wind farm can be seen in the next section (Figure 13), where flow is first visualized for two cases.

### 4.2 Results

The constrained Mann-model turbulence fields (discussed in Section 3) were used to drive LES of each member in the ensemble of eight cases, including aero-elastic calculations for the nine turbines of the simulated farm via the Flex5 coupling.

### 4.2.1 Analyses of cases and comparison with stand-alone simulations

We start by considering the results from the large-eddy simulations for the first (upwind) turbine. This is done to evaluate the
parameter space represented by the cases simulated with LES, in comparison to the stand-alone Flex5 simulations discussed in the previous section; this is because the forcing technique is expected to potentially modify the ramp characteristics (but not substantially change the ensemble's utility in representing the parameter space shown previously in Figure 5). To check the ramps themselves, Figure 11 shows the wind speeds at hub-height for the first turbine; these are taken from the Flex5 channel which reports speed at a distance of *R*=40m upwind.


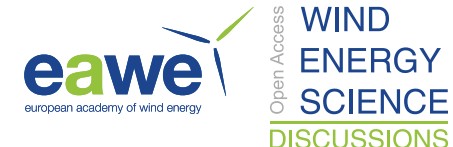

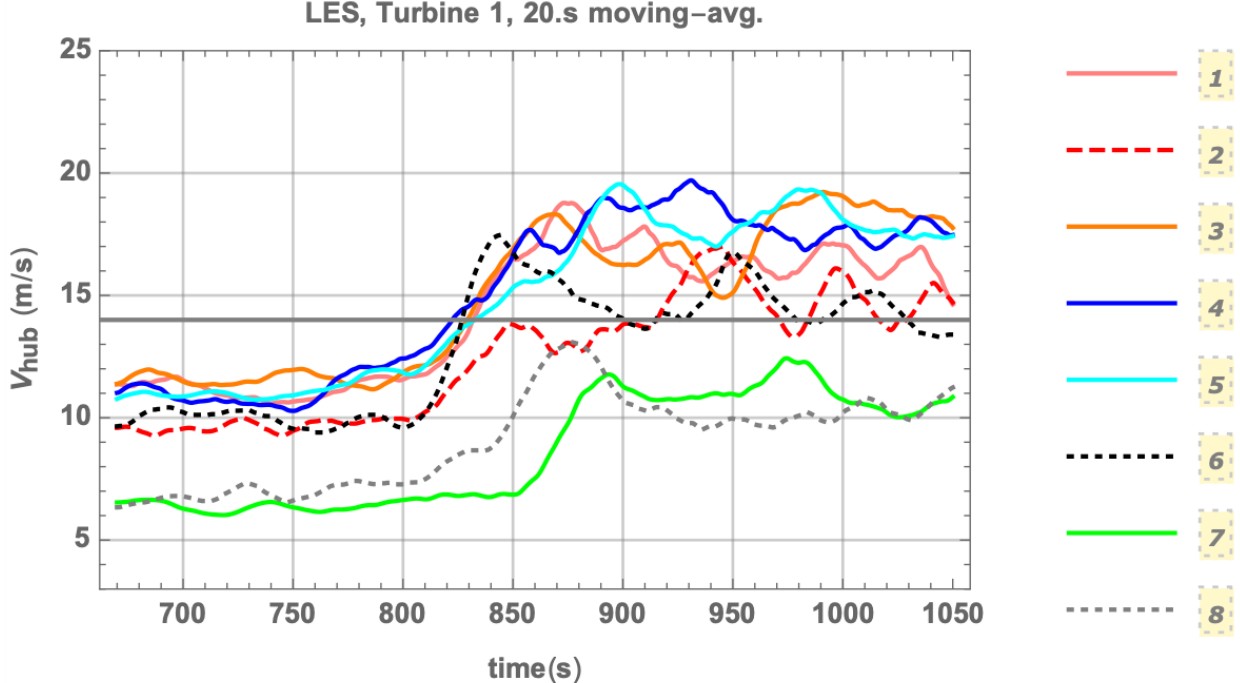

**Figure 11. Wind speed ramps from LES driven by Mann-model simulation, for upwind/first turbine. (Compare to stand-alone Flex5 simulations driven by Mann-model, shown in Figure 9.)**

In comparison with Figure 9 for the stand-alone Flex5-model driven by constrained Mann-model turbulence including the ramps, one can see in Figure 11 for the LES that for most cases the ramp amplitudes are somewhat damped (~10–30%), and some of the cases have $\Delta U/\Delta t$ 'swapped'. However, the ramp accelerations are not appreciably affected: essentially the same space of $\Delta U/\Delta t$ is covered by *the ensemble,* both as planned and simulated in the stand-alone runs. For the downwind ('waked') turbines, Figure 12 shows the wind speed ramps in the LES at turbines 2, 4, 8, and 9.


**Figure 12. Wind speed ramps at hub-height in LES driven by constrained Mann-model simulation. Upper-left: at turbine 2; upper-right: at 4th turbine; lower-left: at 8th turbine; lower-right: at 9th turbine. Colour/case legend follows Figure 11.**

The progressive damping of wind speeds at increased distances within the simulated farm is evident in Figure 12, which shows the wind speeds at all turbines (for more plots see also Fig. 27/Appendix of Kelly *et al.*, 2019b). The figure also demonstrates that as the speeds diminish due to the wake effect, for various cases the wind speed does not exceed $V_{rated}$ in parts of the wind farm's interior. Higher loads within the farm are expected when this happens, such as for case 4 between turbines 8 and 9 (blue line in bottom plots of Figure 12). The ramps remain relatively intact, particularly for higher wind speeds. The maximum

ramp accelerations simply persist for shorter times due to wake turbulence; i.e., 'wiggles are added' to the timeseries $U(t)$ as a ramp progresses through the farm.





The behaviour of ramp events propagating through a wind farm is illustrated in Figure 13, which displays a snapshot of the wind speed field in the farm simulated by LES; cases 6 and 7 are shown at the same time after simulation start. In the figure

it is evident that the ramp has not travelled as far through the farm in case 7 as in case 6 due to the latter case having higher speeds (note the colour scale is different in the two plots); in case 7 the mean flow at hub height does not exceed $V_{rated}$ within the wind farm. In case 6 the flow at hub height ceases to exceed $V_{rated}$ between turbines 3 and 4, causing the ramp-affected loads to be different downwind of turbine 3; there the loads become a bit more consistently higher after the ramp, compared to turbine 3.


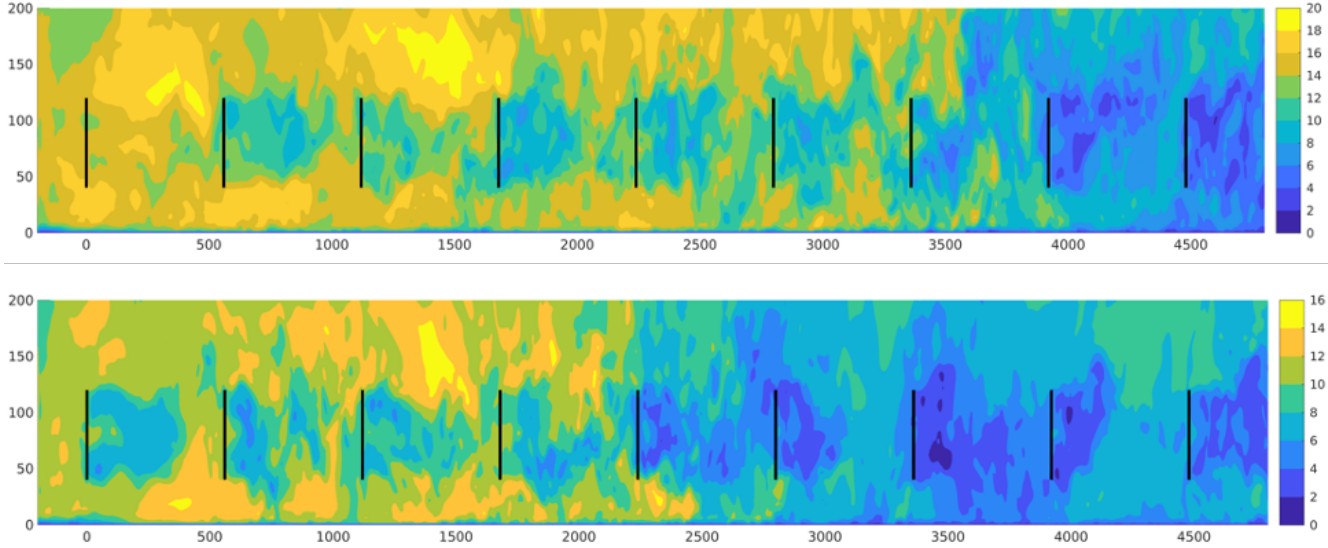

**Figure 13.  Cross-section of wind fields from LES, taken along turbine center (y=0): cases 6 (top) and 7 (bottom), shown at the same time after start.  Vertical axis is height above the surface [m], horizontal axis is streamwise position [m], and flow travels from left**
**to right.  Black lines indicate turbine rotor positions, and color is streamwise wind velocity component in m s⁻¹.**

The predicted power tracks the wind speed up to rated power. This is shown for both the stand-alone (CMM/Flex5) and upwind turbine of the large-eddy simulation model-chain in Figure 14. Similar simple behavior occurs for all downwind turbines in the LES, and is thus not investigated further in this work.[7]


---

[7] The power for all downwind turbines can be seen in the Appendix of the technical report by Kelly *et al.* (2019b).




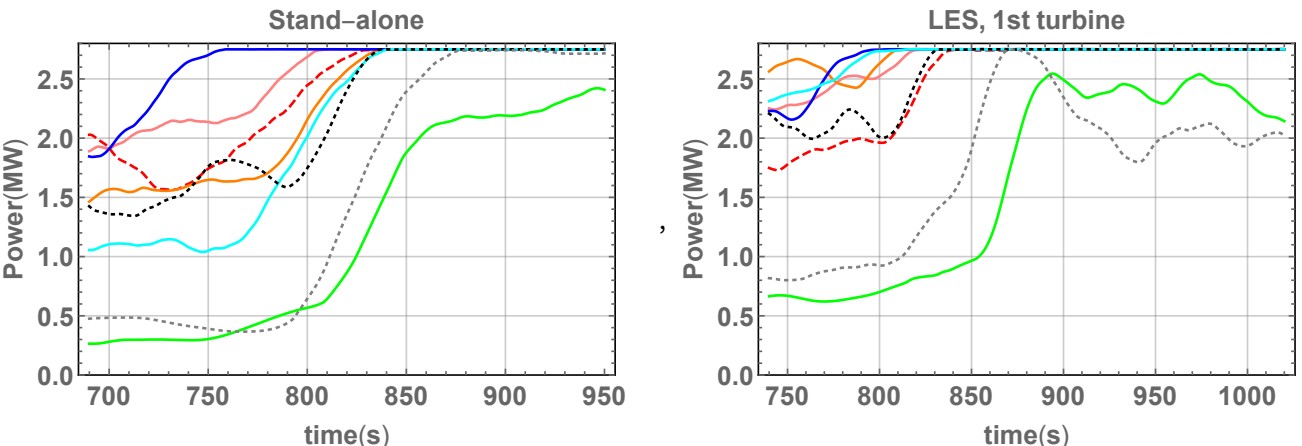

**Figure 14. Power output through all simulated ramp events of ensemble. Left: stand-alone Flex5 simulations driven by CMM. Right: CMM-LES-Flex5 simulated power, for first (upwind) turbine.**

Figure 15 shows tower-base fore-aft moments over passage of the ramps for both the stand-alone CMM/Flex5 simulations and first turbine in the large-eddy simulations, normalized by the respective pre-ramp values for each case. Figure 16 further shows these dimensionless loads for two downwind turbines (numbers 2 and 5) from the LES.

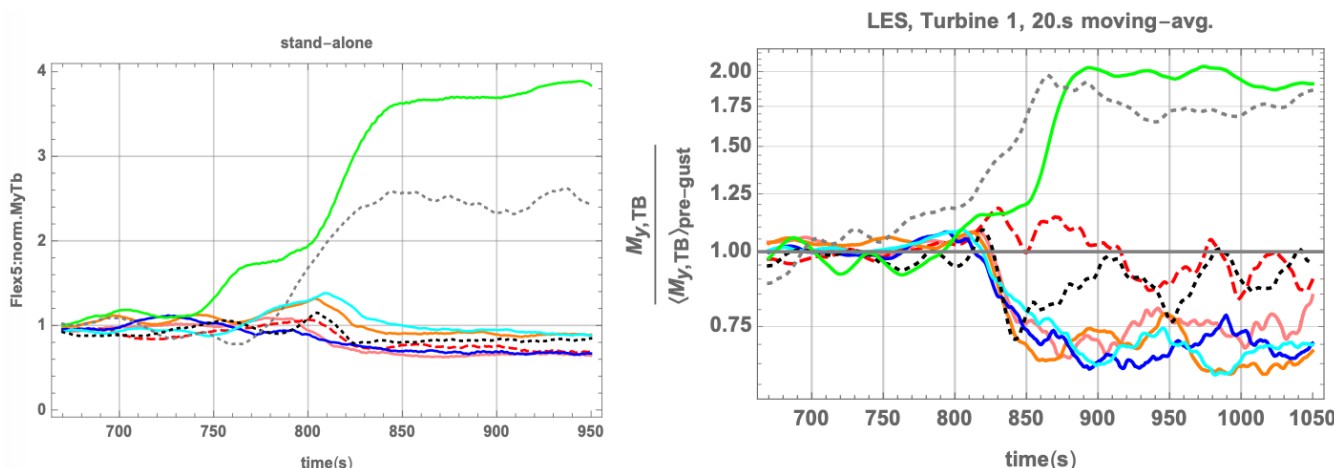

**Figure 15. Tower-base fore-aft moment $M_{tbfa}$, normalized by pre-ramp mean values, for all 8 ramp cases (colors same as in Figure 11). Left: stand-alone Mann-model/Flex5 simulation; right: coupled LES, first (upwind) turbine.**

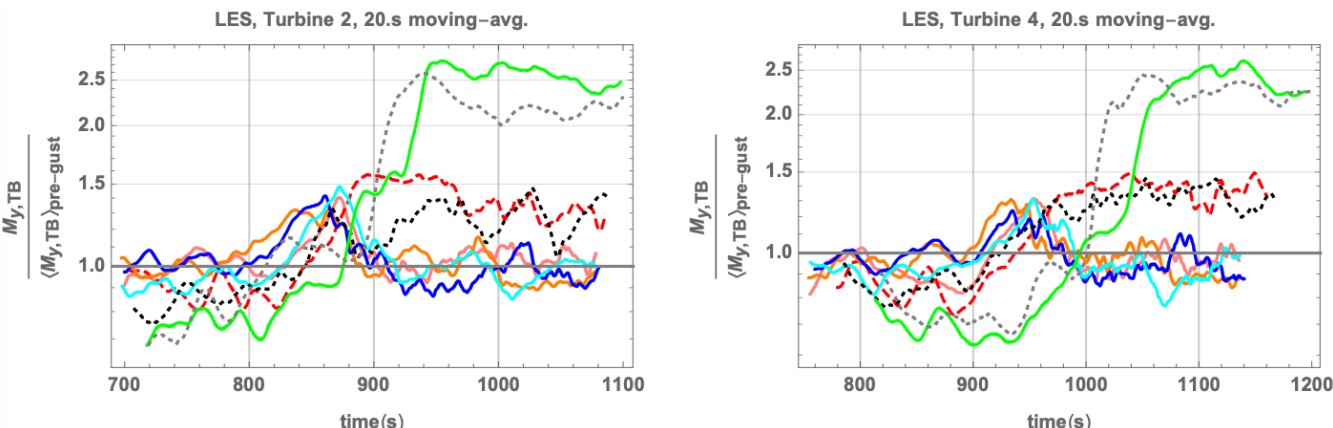

**Figure 16.** Tower-base fore-aft moment $M_{\mathrm{tbfa}}$ normalized by pre-ramp mean values (as in Figure 15b), for 2$^{\mathrm{nd}}$ and 4$^{\mathrm{th}}$ turbines in LES of wind farm. Colour/case legend follows Figure 11-12, 14-15.

From Figure 15–Figure 16 we note several trends. The tower-base loads, relative to their pre-ramp values, exhibit two simple behaviors: for cases in which rated speed is exceeded, the maximum load occurs during the ramp before $V_{\mathrm{rated}}$ is reached; in cases not exceeding $V_{\mathrm{rated}}$ the maximum loads simply correspond to maximum wind speeds attained. For the turbines *not* in the wake of others (i.e., stand-alone or 1$^{\mathrm{st}}$ row of LES), in cases where $U$ exceeds $V_{\mathrm{rated}}$ the peak loads are essentially the same per given acceleration. For cases *not* attaining rated power, the LES loads are lower due to lower overall mean wind speed. In terms of physical (not relative) tower-base loads, ramps not exceeding rated speed have lower overall loads; this is shown in Figure 17.

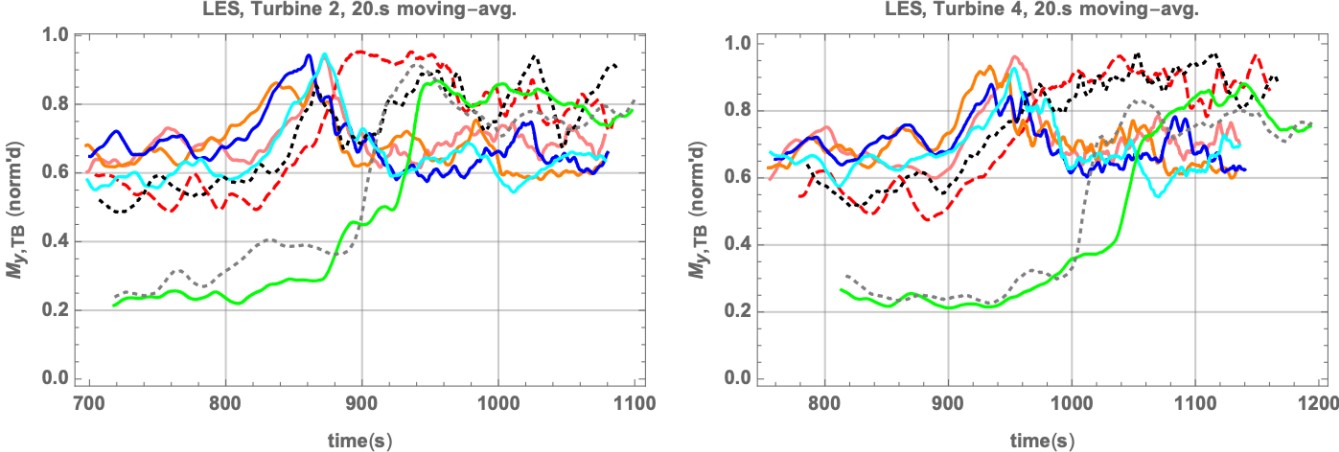

**Figure 17.** "Raw" tower-base fore-aft moment $M_{\mathrm{tbfa}}$ for turbines 2 and 4; note for commercial/proprietary reasons, all values have been non-dimensionalized by a single constant. Colour/case legend as in Figure 11-12, 14-16, 18-19, 22-23.





Figure 18 shows the largest of the 3 blades flap-wise root bending moments for each case; these exhibit similar behavior as the tower base fore-aft moments considered above. Again, some differences exist between the stand-alone CMM/Flex5
simulations and the coupled LES due to the slightly weaker mean wind speeds in the LES, but the peak loads for cases crossing $V_{\text{rated}}$ are notably similar per ramp acceleration. The blades' flap-wise loads for downwind, i.e., waked, turbines exhibit the same behaviour (not shown; see Fig. 29 of Kelly *et al.*, 2019b). Figure 19 further displays the *relative* flap-wise bending moments, i.e., $M_{\text{brfw}}$ normalized by the respective pre-ramp mean values (again for the stand-alone and unwaked LES results); it more clearly demonstrates the similarities between cases exceeding rated speed, similar to the peak tower base loads $M_{\text{tbfa}}$.


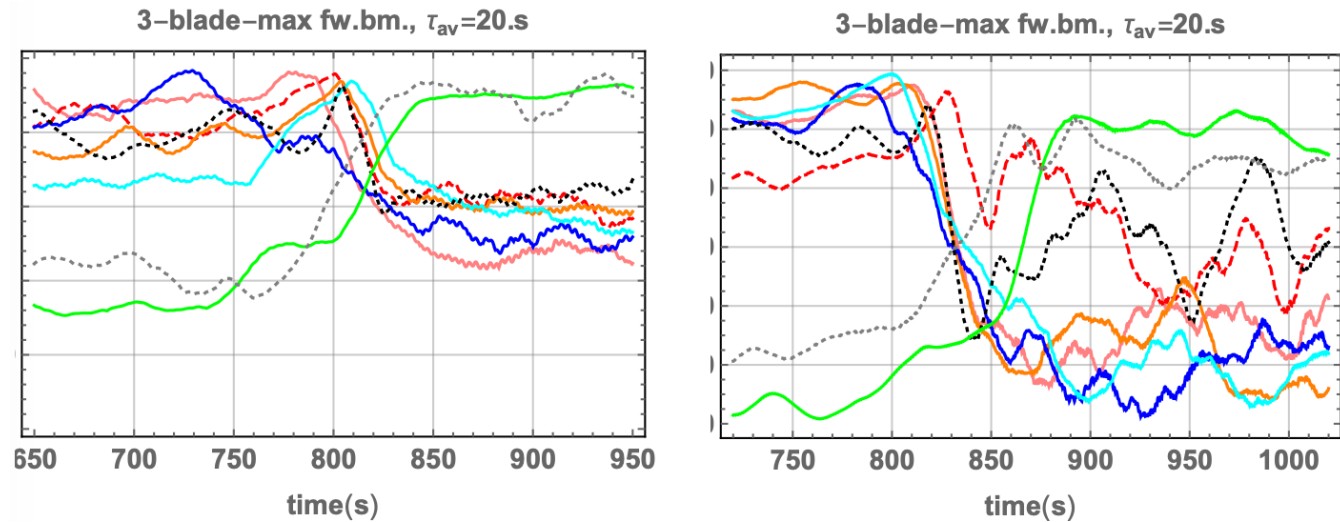

**Figure 18. Maximum flap-wise blade root bending moment $M_{\text{brfw}}$ over all 3 blades. Left: stand-alone Flex5 simulations driven by constrained Mann-model (CMM). Right: coupled LES outputs, for first (upwind) turbine. Dimensional results shown on same axes; values hidden to protect proprietary information. Colour/case legend as in Figure 11-12, 14-17, 19, 22-23.**





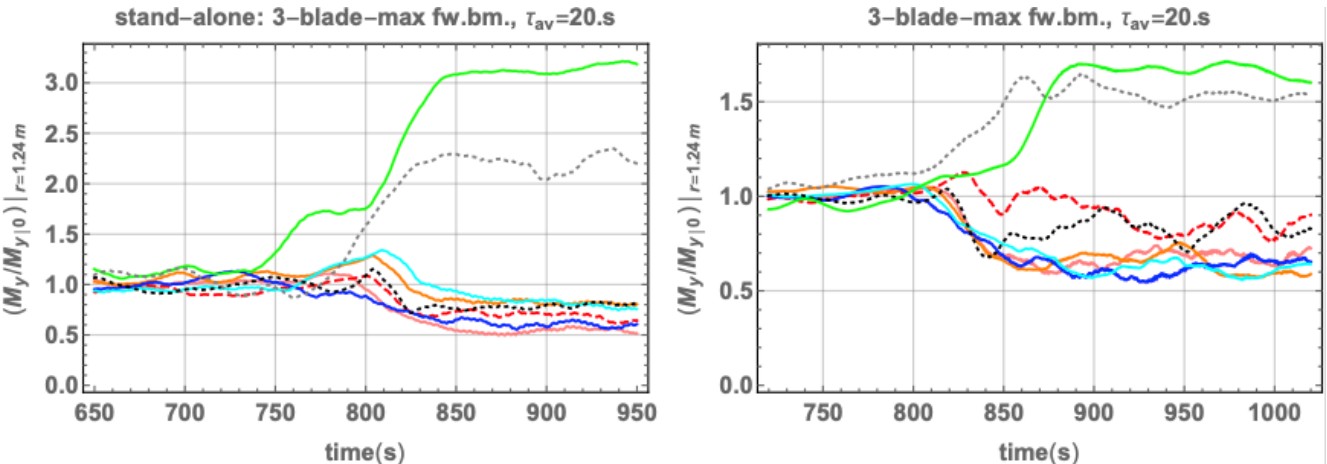

**Figure 19. Flap-wise root bending moment $M_{\mathrm{brfw}}$, maximum of 3 blades; normalized by pre-ramp values. Left:** *stand-alone* **CMM/Flex5 simulations. Right: coupled LES outputs, for** _**first (upwind) turbine**_**. Colour legend as in Figure 11-12, 14-18, 22-23.**


The behavior of ramp-affected $M_{\mathrm{brfw}}$ does not vary much from blade to blade in the stand-alone simulations (<~3%); this is shown in Figure 20, and such variation is even smaller for the corresponding (upwind) turbines in the LES (not shown, see Fig. 4.9 of Kelly *et al.*, 2019b). For the peak loads, including those which are ramp-induced, the same behaviour is exhibited by both the LES and stand-alone model-chains, including the waked turbines in the LES: peaks for ramps exceeding $V_{\mathrm{rated}}$

occur during the ramp, and are dependent upon the ramp-associated $\Delta U/\Delta t$, while load peaks for weaker ramp events (not exceeding rated speed) are determined more by the maximum wind speed. In general, the behavior of both dimensional and normalized loads seen above for single or upwind turbines is also exhibited for downwind turbines in the LES; the blade-root and tower-base loads depend primarily upon the ramp acceleration, for the cases in which rated power is achieved during the ramp event.






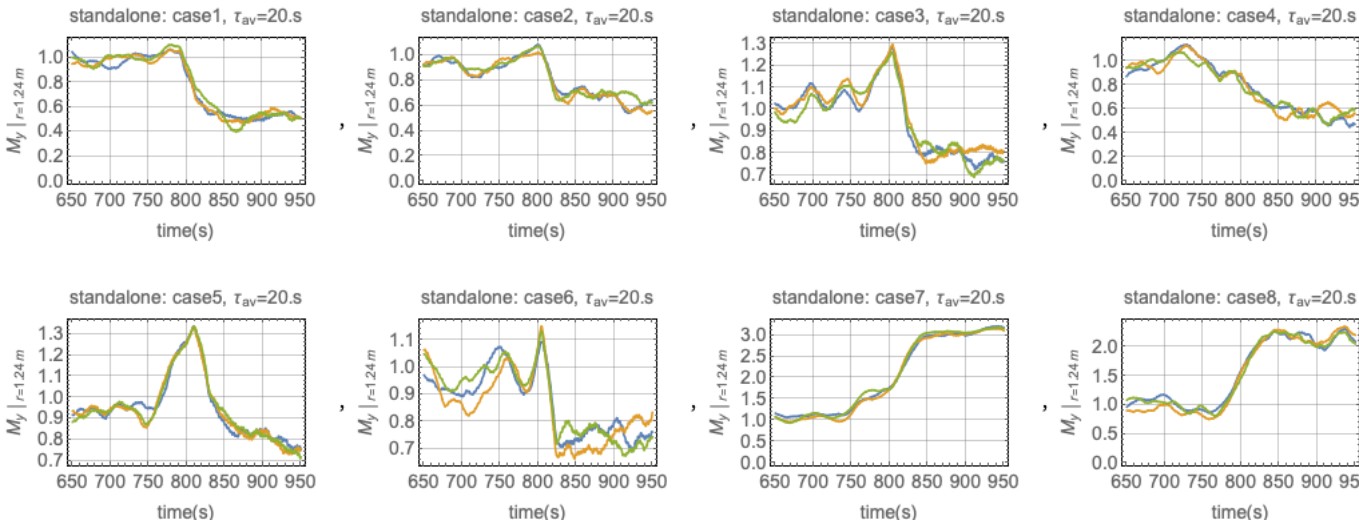

**Figure 20. (4.8) Stand-alone CMM/Flex5 simulations: normalized blade root flap-wise bending moment [$M_{brfw}$] near base ($r$ =1.24m) for each blade (blue/gold/green); loads are normalized by respective pre-ramp mean value.**

455 Maxima of $M_{brfw}$ are shown versus bulk ramp accelerations in Figure 21, for turbines 3 and 8 in the coupled LES. The bulk accelerations are calculated directly using the before-after wind speed difference $\Delta U$ divided by the ramp's rise time $\Delta t$. We remind that the accelerations (and to a lesser extent the loads) obtained can depend on the averaging time, particularly in the stand-alone (CMM-Flex5) simulations. From Figure 21 one can see the loads following proportionally to $\Delta U / \Delta t$, particularly for turbine 3. Further downwind, where the ramps begin failing to exceed rated power, the wakes can add noise to this picture,

460 as demonstrated for turbine 8 in the figure. However, in the coupled LES the *sensitivity* of loads to acceleration (slope of the plots in the figure) remains relatively constant progressing into the farm, though we note this corresponds to an increase as percentage of the loads. As the post-ramp speed begins to not exceed $V_{rated}$ deeper into the farm, then (wake) turbulence can tend to cause the highest loads, instead of ramp accelerations. The sensitivity of the flap-wise blade root bending moments and the tower-base fore-aft moments in the LES model-chain is essentially equal to that found in the stand-alone simulations:

465 again ~3% of $M_{brfw}$ or $M_{tbfa}$ per 0.1 m·s$^{-2}$ of $\Delta U / \Delta t$, though it varies, as is shown below.





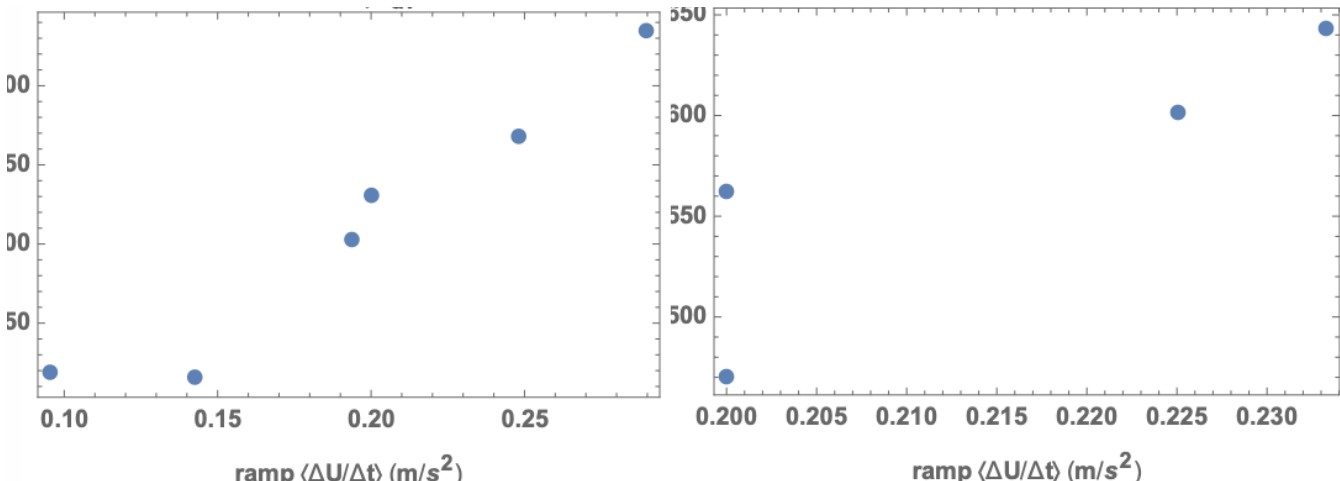

**Figure 21. Flap-wise blade root bending moment $M_{\mathrm{brfw}}$ (max. of 3 blades) versus ramp bulk acceleration, for cases with ramp-induced $U > V_{\mathrm{rated}}$: turbine 3 (left) and turbine 8 (right), from LES/Flex. Vertical-axis values obscured for proprietary reasons.**


As long as rated power is achieved, the loads during the ramp events are relatively constant into the farm ($M_{\mathrm{brfw}}$ and $M_{\mathrm{tbfa}}$ vary $<\sim$10%), with the exception of the first two turbines (due to the wake first arising). This is demonstrated for the root flap-wise bending moments of the blades in Figure 22, which shows the mean $M_{\mathrm{brfw}}$ over all three blades during ramp passage at each turbine. For cases with wind speeds not exceeding $V_{\mathrm{rated}}$, the loads decrease progressing further into the farm. For these sub-

rated cases (6 and 7) the right-hand plot also shows how the loads first grow relative to their pre-ramp values, then decay as the ramp propagates through the farm and $U$ decreases. The left-hand plot in Figure 22 is consistent with the ramp accelerations remaining undamped from turbine to turbine, while the right-hand further shows that for all cases the ramps tend to increase loads after the first turbine, at least due to the associated increase in wind speeds (with subsequently reduced $C_T$) and wake. Nearly identical results arise when considering the normalized tower base fore-aft moments $M_{\mathrm{tbfa}}$ (not shown). In Figure 22

one can also see that when rated power is achieved, the dominant blade loading is relatively constant throughout the wind farm; when $V_{\mathrm{rated}}$ is not exceeded, then the loads tend to decrease within the farm along with the speed. One can see within the farm where $U > V_{\mathrm{rated}}$ that the rotor-mean loads do not differ much from their pre-ramp values; however, in some cases the speed falls below rated (e.g. case 2 downwind of turbine 1) and the loads are higher than their pre-ramp values.





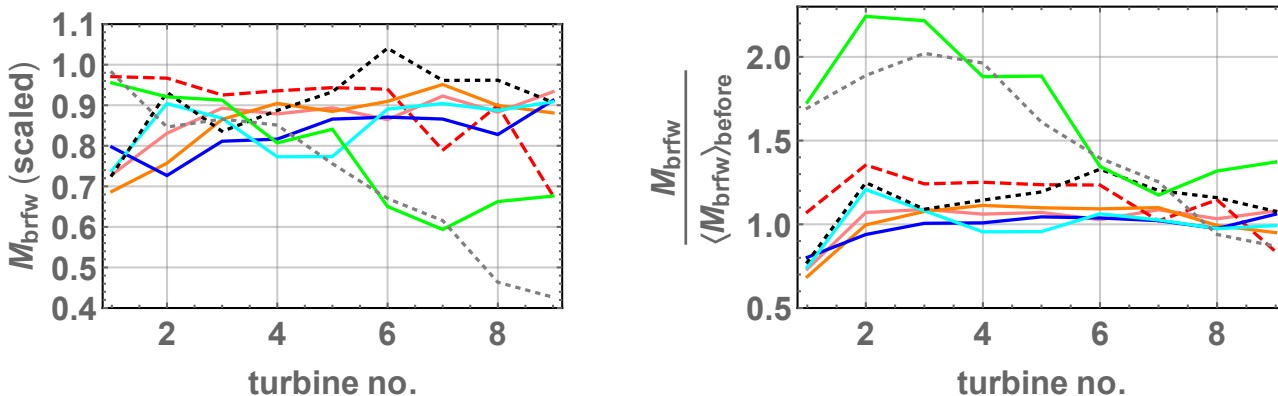


**Figure 22. Mean flap-wise blade root bending moment $M_{brfw}$ during ramp passage for all cases (averaged over all blades), versus distance into the farm (recalling turbine separation of $7D = 560m$). Left: in arbitrary units (proprietary); right: normalized by pre-ramp values. Colour/case legend same as Figure 11-12, 14-19, 23.**

The maximum simulated ramp-induced loads found at each turbine behave similarly to the mean values shown above in terms of evolution through the farm, but not in terms of magnitude. Figure 23 shows the maximum flap-wise root bending moment versus turbine number for each case simulation. Relative to their values before the ramps, the loads are significantly larger, particularly within the wind farm. The ramps are seen to increase the *maximum* loads, even as $V_{rated}$ is exceeded and obviously when $V_{rated}$ is not exceeded, since the wind speed is simply higher. This is also the case for the tower base fore-aft moments.


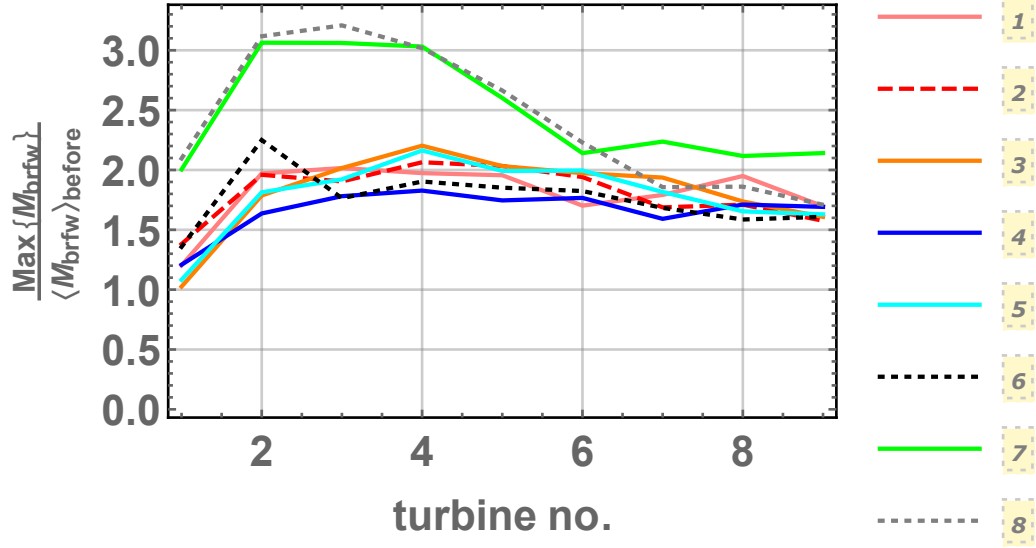

**Figure 23. Maximum flap-wise blade root bending moment $M_{brfw}$ for all ramp cases from coupled LES, as function of distance into the farm (turbine number, where turbine separation is 560 m). Colour/case legend same as in Figure 11-12, 14-19,22.**



### 4.3 Ensemble ramp parameter space and effect on loads

Thus far we have discussed the effects of two of the three input parameters defining the wind ramp simulation ensemble, i.e., ramp acceleration and wind speed. The input space also included shear above hub-height, in order to address the variable amount of entrainment expected, and its effect on the flow and wind farm during ramp events. The pre-ramp shear in some cases were shifted in the full model-chain, compared to ensemble prescription (Table 1). In case 5 the LES gave an upwind shear before the ramp of $-0.005$ s$^{-1}$ instead of 0, and the nonzero (positive) shear cases (1,2,4,6,8) had inflow $dU/dz$ diagnosed to be approximately half the design choice. Regardless of its magnitude, the upwind pre-ramp shear was not found to directly impact the shear or entrainment above and within the simulated farm, and the LES fields exhibited little correlation between $dU/dz$ upwind and within/over the farm; this is consistent with a 'windfarm boundary-layer' developing (e.g. Porté-Agel *et al.*, 2020). Before ramp passage, the cases with lowest speeds have smaller $dU/dz$ between turbines, but after passage, the cases with lowest wind speed tend to exhibit higher upper-rotor shear in the farm, especially downwind of the first three turbines. Both before and after ramp passage, within the farm $dU/dz$ over the upper half-rotor ranges from 0 to 0.16 s$^{-1}$ (up to ~8 times ambient values) due to the turbines, with the difference of $dU/dz$ being positive or negative depending on distance downstream and case. There is no straightforward relationship between shear before and after ramp passage at these heights, as with the ambient-condition observations; thus we do not include plots of the shear here (Kelly *et al.*, 2019b include such).

Returning to the effects of ramps on loads, we note the ramp accelerations shown earlier with loads (e.g. Figure 21) were calculated in bulk, as the ratio of ramp amplitude to duration, $\Delta U/\Delta t$. But if one considers the accelerations calculated directly at each timestep of the simulated timeseries (every 0.04 s), then its relation to loads becomes difficult to see, especially if $dU/dt$ is only considered at a single point in space. Using different averaging times (e.g. from 5–60s), a correlation between loads and $dU/dt$ appears, but the optimal averaging time depends on the ramp duration, and such dependence was not clear; more work involving filtering and turbine response would better clarify this.[8] The accelerations are dependent on the averaging time used, and we remind that turbines act as low-pass filters, responding less to shorter duration accelerations. The spectral response of wind turbines—as well as their control systems—is beyond the scope of this work, though it can be considered relative to the spectral content of accelerations. Given all this, and that we are using only a particular control system and turbine, we have focused on response to the bulk acceleration as above.

For the tower base fore-aft and blade root flap-wise bending moments considered here, the maximum loads for each turbine and case occur during the ramp events, near the time when wind speed crosses $V_{\text{rated}}$. The maximum of tower base fore-aft

---

[8] Note that different turbines, as well as different control systems/strategies, will have markedly different spectral responses; thus an optimal averaging time is not certain nor trivial.





moment ($M_{\text{tbfa,peak}}$) and blade root flap-wise bending moment ($M_{\text{brfw,peak}}$) are generally correlated to maximum acceleration
for single (unwaked) turbines, as previously shown by e.g. Hannesdóttir *et al.*, 2019. But in a windfarm the peak accelerations
can be comprised of both the turbulence (primarily from the wake) as well as the ramp acceleration. Short-duration
accelerations (with characteristic timescales much smaller than the ramp rise-time $\Delta t$) due to the wake are not directly relatable
to the ramp acceleration, and can be ~10–30 times larger than the the *bulk* acceleration $\Delta U/\Delta t$. Thus when plotting maximum
load versus maximum acceleration, there is some 'noise' in addition to the trend that one sees. This is shown in Figure 24,
which includes all turbine responses where a ramp causes wind speed to exceed $V_{\text{rated}}$. The sensitivity of peak loads to *peak*
acceleration appears to be an order of magnitude smaller than the sensitivity to *bulk* ramp acceleration: $\partial M_{\text{brfw,peak}}/$
$\partial(\mathrm{d}U/\mathrm{d}t)_{\text{peak}} \approx 6\%$ per m·s$^{-2}$ and $\partial M_{\text{tbfa,peak}}/\partial(\mathrm{d}U/\mathrm{d}t)_{\text{peak}} \approx 3\%$ per m·s$^{-2}$. However, this is again via single-point
calculations which are not representative of the whole rotor or turbine, and partly due to wake-related accelerations
uncorrelated to the ramp.


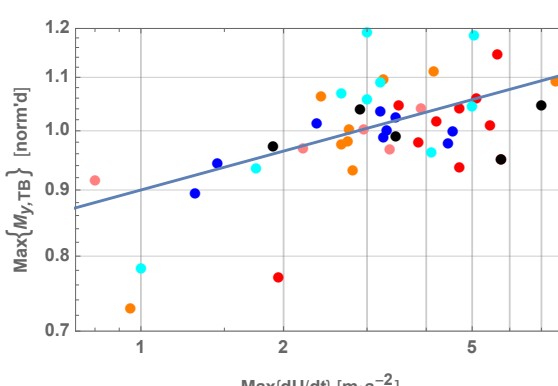
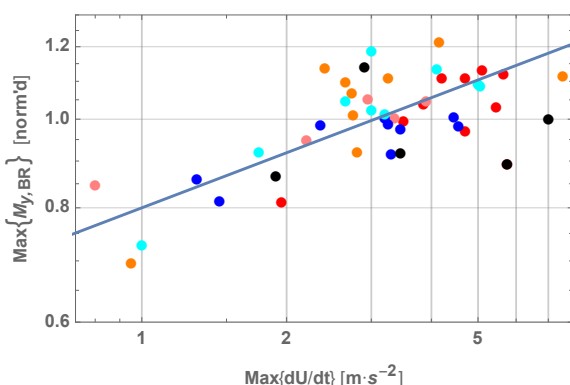

**Figure 24. Maximum loads ($M_{\text{tbfa}}$ at left, $M_{\text{brfw}}$ at right) versus maximum acceleration: all cases and turbines, where speed exceeds rated. Loads values normalized by mean of all points to protect proprietary information. Colors correspond to cases following previous figures. Lines indicate sensitivity: log-log slope of 0.1 at left, 0.2 at right.**


For context, we revisit the sensitivity of flap-wise blade root bending moment $M_{\text{brfw}}$ and tower-base fore-aft moment $M_{\text{tbfa}}$ to
*bulk* ramp acceleration $\Delta U/\Delta t$: $\partial \ln M_{\text{brfw}}/\partial(\Delta U/\Delta t)$ and $\partial \ln M_{\text{tbfa}}/\partial(\Delta U/\Delta t)$ were seen to be the same in both stand-alone
and LES coupled simulations (expressed as percentage change in load per acceleration). This is not completely unexpected,
recalling that both load types are directly driven by the thrust force. Perhaps more importantly, the loads are increased by
roughly 45% for $M_{\text{brfw}}$ and 50% for $M_{\text{tbfa}}$ relative to their values during the non-ramp conditions simulated (Figure 10); this
is also true for the unwaked turbines in the LES. For the NM80 turbine and its control system used here it is expected that the
$M_{\text{brfw}}$ and $M_{\text{tbfa}}$ for un-waked turbines will vary by ~45% or 50% respectively, plus ~3% per 0.1 m/s$^2$ of $\Delta U/\Delta t$ based on the
simulated ensemble derived from a decade of observed ramps. Thus the maximum observed ramp acceleration of 1 m/s$^2$ from





the 11 years of data could correspond to an increase in loads of ~75% or more relative to nominal conditions. The sensitivity to ramp acceleration is also found to be the same for downwind turbines in wakes (implicit in Figure 21), though we remind that the bulk ramp accelerations can differ from the upwind 'incoming' ramp $\Delta U / \Delta t$. We note that the range of $\Delta U / \Delta t$ in the ensemble (Table 2) roughly corresponds to ramps expected to occur about twice per year offshore, though weaker ramps obviously occur more frequently (Figure 2). The ramp parameter distributions shown in Section 2 and the results given in sections 3–4 are also consistent with the rate at which offshore ramp events are expected to give conditions exceeding the IEC 61400-1 standard in terms of 10-minute standard deviations of wind speed, following the findings of Hannesdóttir *et al.* (2019).

**5 Conclusions**

The statistics of wind speed ramps, and their effects on loads within a wind farm—including sensitivities of loads to ramp characteristics—have been investigated here. Specifically, we focused on the tower-base fore-aft bending moment and flap-wise blade root bending moment. Such quantitative work was facilitated by [1] statistical reduction based on low-order physics and micrometeorology, via [2] long-term high-frequency observations, along with [3] the incorporation and use of an appropriately coupled high-fidelity model-chain. The coupled models comprising the latter are constrained Mann-model turbulence simulation, large-eddy simulation including an actuator-line model, and the aero-elastic loads model Flex5. Through two model-chains of coupled simulations (Mann-model to Flex5, and Mann-model to LES with actuator line modelling and Flex5), using a statistically representative ensemble of cases based on the reduced-parameter probability space derived from effectively offshore observations, we were able to find and explain a number of effects. The main results are summarized in the list below.

- A compact distribution of relevant parameters describing wind speed ramps in offshore conditions was found, based on long-term observations and accounting for the dominant physics.
- Ramps causing the wind to exceed rated speed ($U > V_\text{rated}$) offer the highest maxima of blade-root flap-wise bending moment and tower base fore-aft moment; in these cases the load maxima depend primarily on ramp acceleration.
- For ramps that do not exceed rated speed ($U < V_\text{rated}$), the loads depend on $U$ more than on ramp acceleration; however, in these cases the speed and turbulence *combined* can result in load maxima.
- The bulk ramp accelerations $\Delta U / \Delta t$ persist through the farm, despite generation of wake turbulence and the decrease of mean winds into the farm.
  - Downwind of the second turbine, if/where rated speed is exceeded, ramp-associated peak loads are relatively constant through the farm.
- As mean wind decreases further into the farm, ramps can begin to have $U < V_\text{rated}$, leading to higher loads relative to pre-ramp values.
  - The distance into farm where loads begin to increase depends on the ratio $U_\text{post-ramp} / V_\text{rated}$.
- The maxima of blade-root flap-wise bending moments ($M_\text{brfw}$) and tower base fore-aft moments ($M_\text{tbfa}$) each had a sensitivity of roughly 3% per 0.1 m s$^{-2}$ of bulk ramp acceleration for the turbine considered (NM80). This was found in both in stand-alone and coupled LES simulations.



- In un-waked conditions the total increase in $M_{\text{brfw}}$ and $M_{\text{tbfa}}$ due to wind speed ramps was 45% and 50%
respectively, plus 3% per 0.1 m s$^{-2}$ of $\Delta U/\Delta t$ for the turbine considered.

With regard to further work and improvements, we note several things. The simulated flows lack the effect of stability, particularly the stable capping-inversion, which acts to maintain shear above the farm while moderating entrainment into the farm from above. LES of such events through wind farms can incorporate different capping-inversion strengths and inversion
heights, in order to investigate the effect of the shear as well as the inversion height and strength. However, we note the rarity of ABL depths below 300 m (Liu & Liang, 2010), i.e., twice the upper rotor tip of the wind turbines simulated here. The wind ramp results are not expected to be significantly affected by such inclusion of stable capping inversions in most cases, though future analysis of both observations and simulations can address this issue. We also note the results here were obtained using wind turbines with a rated speed of 14 m/s, whereas $V_{\text{rated}}$ of 12–13 m/s is commonly seen. Lower $V_{\text{rated}}$ will increase the
occurrence of ramp-affected speeds crossing into the rated power regime, but the joint distributions found here do not change significantly when considering different $V_{\text{rated}}$; the sensitivity of loads to the ramp accelerations will not necessarily change either. However, for different turbines and control systems (or operational regimes), one can expect different sensitivity to ramp events; the statistics and findings here, can be useful to estimate expected impacts upon wind farms with other turbines.

## Author contributions

M.K. conceived the statistically driven model chain and sensitivity concepts, performed the statistical analysis and probabilistic characterization, derived the case ensemble, analysed and visualized the results, and wrote the manuscript draft. S.J.A. performed the coupled large-eddy/actuator-line/aeroelastic simulations, designed the LES setup (with M.K.), designed and implemented Mann-model forcing in the large-eddy simulation code (Ellipsys), and wrote the basis of section 4.1. Á.H. executed her ramp-detection algorithm to identify observed ramps and retrieve ramp parameters and ran the constrained
Mann-model simulations for input into Flex5 and LES. All authors have contributed to correcting the manuscript.

## Competing interests

The authors have no competing interests to declare involving this work.

## Acknowledgements

We thank Hans Ejsing Jørgensen for informing of Carbon-Trust's Offshore Wind Accelerator funding call and connecting with the Fraser-Nash Consultancy, whom helped to secure funding for (and administrate) this project; also thanks to Niels N. Sørensen for input on the numerical methodology.



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
