# Peer review of "Statistical impact of wind-speed ramp events on turbines, via observations and coupled fluid-dynamic and aeroelastic simulations"

_Wind Energy Science, 2021_

## Author Response (AR1)

Thanks to the reviewers for their constructive comments.

First, we responded to three comments issued by the reviewer in RC1:

1. As mentioned in the second paragraph of section 2/footnote 2, this turbine was chosen because the original study was designed for comparison with the offshore Rødsand 2 windfarm, which employs a very similar turbine and controller (Siemens 2.3MW). The latter was not available due to proprietary reasons, and the NM80 has been previously used for this purpose.

2. Good point. The ramp events are not simply modelled as normal wind fields, but as background turbulent flow plus event, via constraints. As written in the draft/to repeat: the vertical shear is controlled per the observed event space, with zero horizontal shear and veer, and the the turbulence length scales corresponding to the background flow; the coherences of the background turbulence are dictated by the Mann model/LES. I also remind that we mention in the draft that the simulated events were taken to have no vertical tilt, though this can occur (Hannesdóttir *et al.*, 2019).
The spatial characteristics of the ramp events were not the focus of this work, though our previous student (Alcayaga [2017], referenced in the draft) did examine their vertical structure and turbulence around them. In the 2017 study it was confirmed that the TKE and anisotropy during events is increased, and the length scale decreased, as postulated by *Hunt et al.* (2010). However, these effects are not dealt with here: $\sigma_u$ is dominated by the ramp amplitude, so our results are slightly conservative. The change in anisotropy negligibly affects the streamwise component considered (c.f. Kelly, 2018), as does the smaller effective length scale during events. However, these are subject to further research.

3. The earlier works of Hannesdóttir & Kelly (2019) and Hannesdóttir, Kelly, & Dimitrov (2019) dealt with this aspect; the former showed that the rise times of most ramp events are longer than prescribed in the IEC standard, while the latter found that tower-base fore-aft moments can exceed the standard's DLC 1.3 for ramps crossing rated speed. The current paper follows after this: now we focus on how the ramps travel (persist) through wind farms, the sensitivity of thrust-dominated loads to the ramps (for single turbines or within farms), and the statistical implications of such, given the ramp joint probability space.
Regarding fatigue loads, their assessment (e.g. per bin) is a current topic being researched now e.g. in the Hiperwind project (https://www.hiperwind.eu/).

In response to the "minor issues" in RC2, we replied:

- Following the reviewer's suggestion, we have added a figure and short "walk-through" for one case.
- Regarding the comment about the conclusion sub-point (line 583-585):
the major bullet-point above this sub-point already states that $U<V_{rated}$ and it is already understood that $U_{post-ramp}>U_{pre-ramp}$. The reviewer's statement about shorter distance downstream is not necessarily correct; we wrote about the basic dependence on the ratio $U_{post-ramp}/V_{rated}$ because it can depend on the wake situation and does not necessarily correlate with distance into the farm. However, given the reviewer's input, we have changed the wording of the major bullet point above this to

specifically include "(again crossing rated speed)", and also changed the text in this sub-point to include "where this happens", referring to the major-point in line 582.

In response to the "comments" offered by the reviewer, we respond:

- These events are offshore, mostly associated with (cold) frontal passages—i.e. the advective transition across a "line" seen on typical weather maps.  Onshore thunderstorms may often be associated with (cold) fronts; however, the accelerations therein tend to be related to downdrafts and local cells, having a different character.
- As shown in Hannesdóttir & Kelly (2019), the amplitude of ramp events does not exceed the IEC's "ECD" prescription for wind speed, but *may do so for directional change*s.
  Regarding the EOG: it is difficult to definitively comment on a direct implication, because the 61400-1's EOG prescription depends on the site-specific extreme and mean speeds ($0.8V_{e50}, V_{hub}$) or turbulence, and imposes a 10.5s rise-time; further, ed.4 of the 61400-1 allows one to replace the analytical "hat" form with stochastic simulations for DLC3.2 (start-up).
- The authors agree that comparison with measurements would be beneficial, and was originally intended in the project, but this was unfortunately not possible.  Doing so would be worth a separate article, but such measurements were not available.  If more manufacturers would share loads measurements over long operational periods (>1y), thus capturing a statistically significant number of load-driving events, then we could certainly find out more.

---

## Author Response (AR2)

**DTU Wind Energy**

**To**    WES / Assoc. Editor Sandrine Aubrun

**Reg.**    Revised submission

17 August 2021

**Reply to Associate Editor's final comment/request**

The associate editor commented that she "noticed several citations of reports that might be difficult to find for the readers. Some are even not written in English. Particularly the citations of two reports written in French (*Ta Phuoc 1994 and Ta Phuoc  et al., 1994 references*). I recommend to make an effort to find more findable references, or if not possible to develop more in details the elements extracted from these reports to be used in the present paper."

This involved two references, cited to describe the LES subgrid/subfilter model as given by its original author.  To remedy this situation, I have added a more recent reference which is more easily obtained (publicly available), that has the equations as implemented in the computational LES model that we employed.

Thanks again for your work and constructive help,

--Mark Kelly, also on behalf of Søren Andersen and Ásta Hannesdóttir